# One Demo Is All It Takes: Planning Domain Derivation with LLMs from A Single Demonstration

**Jinbang Huang[1], Yixin Xiao[1], Zhanguang Zhang[1], Mark Coates[2],**
**Jianye Hao[1], Yingxue Zhang[1]**
[1] Huawei Noah's Ark Lab, [2] McGill University

## Abstract

Pre-trained large language models (LLMs) show promise for robotic task planning but often struggle to guarantee correctness in long-horizon problems. Task and motion planning (TAMP) addresses this by grounding symbolic plans in low-level execution, yet it relies heavily on manually engineered planning domains. To improve long-horizon planning reliability and reduce human intervention, we present Planning Domain Derivation with LLMs (PDDLLM), a framework that automatically induces symbolic predicates and actions directly from demonstration trajectories by combining LLM reasoning with physical simulation roll-outs. Unlike prior domain-inference methods that rely on partially predefined or language descriptions of planning domains, PDDLLM constructs domains with minimal manual domain initialization and automatically integrates them with motion planners to produce executable plans, enhancing long-horizon planning automation. Across 1,200 tasks in nine environments, PDDLLM outperforms six LLM-based planning baselines, achieving at least 20% higher success rates, reduced token costs, and successful deployment on multiple physical robot platforms.

## 1 Introduction

Robotic planning remains challenging in complex scenarios that require abstract, long-horizon reasoning. Large language models (LLMs) demonstrate strong generalization in this domain but often struggle with temporal dependencies in extended tasks (Huang et al., 2022a;b). Task and motion planning (TAMP) frameworks provide robustness in long-horizon reasoning by integrating high-level symbolic reasoning with low-level motion planning. However, they face two major limitations: (i) planning domains, expressed in symbolic planning languages such as PDDL (McDermott et al., 1998), are difficult to ground in complex geometric information, and (ii) these domains are labor-intensive to manually construct (Garrett et al., 2020; 2021; Khodeir et al., 2023; Silver et al., 2021). Recent vision-language-action models have advanced the first challenge by improving geometric grounding through semantic action instructions, but they pay limited attention to long-horizon reasoning (Black et al., 2024; Kim et al., 2024; Zitkovich et al., 2023; Li et al., 2024; Team et al., 2024). Complementarily, our work focuses on solving the second challenge by reducing the human effort required for domain construction, thereby enabling automated and scalable long-horizon reasoning.

Existing domain-generation methods typically rely on predesigned elements (predicates or actions) to complete a domain (Silver et al., 2023; Kumar et al., 2023), where manual predesign remains critical and often depends on curated training data (Huang et al., 2025). LLM-based approaches also demand extensive natural-language descriptions of PDDL domains and careful prompt engineering (Guan et al., 2023; Oswald et al., 2024). Meanwhile, recent advances in world-model learning show that LLMs can serve as compact explainer and reasoner for robot behavior and object relations (Guan et al., 2023; Zhao et al., 2023; Liang et al., 2024; Tang et al., 2024). Motivated by this, we propose to combine LLM reasoning with physical simulation to generate planning domains, eliminating reliance on manual predesign, curated training data, and detailed textual descriptions.

---

Correspondence to: jinbang.huang@h-partners.com, {zhanguang.zhang, yingxue.zhang}@huawei.com

We introduce PDDLLM, an LLM-driven framework for automated planning-domain construction that requires minimal pre-design and no extensive human-provided descriptions. From a single demonstration, the model generate both predicates and actions in a one-shot fashion, yielding an executable planning domain. Moreover, PDDLLM automates integration with low-level motion planners, further reducing the need for manual intervention. Our method targets at long-horizon planning problems, aligning with the scope of TAMP studies. Managing action sequences and task structures is essential for these tasks such as household activities and puzzle solving. The main contributions of this paper are:

- An algorithm that combines LLMs and physical simulation to automatically generate a human-interpretable planning domain from a single demonstration.
- A Logical Constraint Adapter (LoCA) that systematically interfaces the generated domain with low-level motion planners.
- An extensive evaluation on over 1,200 tasks in nine environments, demonstrating superior long-horizon planning performance and more efficient token usage than state-of-the-art baselines, with successful deployment on three physical robot platforms.

## 2 RELATED WORK

**Task planning with pre-trained large models** With the advent of pre-trained large models (LMs), the use of LLMs and vision language models (VLMs) has significantly advanced the performance of task planners (Huang et al., 2022b; Wang et al., 2024; Chen et al., 2024b; Li et al., 2023). Although many studies have demonstrated the generalization capabilities of LM-based task planners, they often lack robustness and struggle with long-horizon tasks that require complex reasoning (Wang et al., 2024; Sermanet et al., 2024; Driess et al., 2023; Ahn et al., 2022; Huang et al., 2022a). To address this limitation, recent research has explored guiding task planners with LM-derived heuristics to accelerate informed search. These approaches integrate symbolic search with LMs to accelerate task planning and reduce search complexity. Notable efforts include heuristics for prioritizing feasible states (Zhao et al., 2023), ranking feasible actions (Yang et al., 2025; Meng et al., 2024; Hu et al., 2023; Zhao et al., 2023), and pruning search trees (Silver et al., 2024). However, a major limitation of these methods lies in their reliance on manually constructed symbolic planning domains to build search trees, which imposes additional development overhead and reduces flexibility.

**Learning planning domains** A recent line of research aims to infer the planning domain directly from human demonstrations, environment interactions, or natural language. However, these approaches often depend on partially or fully predefined symbolic predicates and actions as well as extensive training dataset (Silver et al., 2023; Kumar et al., 2023; Huang et al., 2025; Wong et al., 2023; Liu et al., 2024). Some recent studies have explored leveraging LMs for domain generation, primarily by extending manually defined domains with additional predicates and actions (Liang et al., 2024; Athalye et al., 2024; Byrnes et al., 2025). The approaches proposed by Guan et al. (2023); Han et al. (2024) generates planning domains requiring prompts containing human-crafted planning examples or intense manual feedback.Another line of research investigates generating and refining planning domains through environmental feedback. (Liang et al., 2024; Zhu et al., 2025). Moreover, many studies assume that logical actions have pre-designed motion-level primitive skills (Kumar et al., 2023; Huang et al., 2025; Athalye et al., 2024; Han et al., 2024), requiring labor-intensive alignment between planning domains and low-level motion planners. PDDLLM addresses this limitation by automatically grounding symbolic actions into motion constraints, thereby reducing manual effort and enhancing scalability and autonomy.

## 3 PRELIMINARIES

PDDL is a standard formal language used to specify planning problems. The object set $\mathcal{O}$ represents the environment's objects, whose continuous state $\mathcal{S}$, such as pose, color, and size, can be queried via a perception function $\mathcal{I} : \mathcal{O} \times \mathcal{I} \to \mathcal{S}$. The PDDL domain $\mathcal{D} = (\mathcal{P}, \mathcal{A})$ describes the general rules of the environment, consisting of a set of logical predicates $\mathcal{P}$ and a set of logical actions $\mathcal{A}$. A **logical predicate** $p \in \mathcal{P}$ specifies either intrinsic properties of an object $o$ or relations between objects (e.g., (cooked ?$o_1$), (on ?$o_1$ ?$o_2$)). Each predicate is evaluated by a binary classifier

over the continuous state, returning true or false. A symbolic description of the environment $\mathcal{X}$ can be obtained by grounding $\mathcal{P}$ across $\mathcal{S}$ (i.e. $\mathcal{S} \times \mathcal{P} \to \mathcal{X}$). A **logical action** $a \in \mathcal{A}$ consists of a precondition $\mathcal{P}_{pre} = \langle p_1, p_2, \dots \rangle$ and an action's effect $\mathcal{P}_{eff} = \langle p'_1, p'_2, \dots \rangle$. The precondition represents a set of predicates that must be satisfied for the action to be executed, while the action's effect describes the change of the resultant state upon action completion. Logical actions define the logical state transitions $\mathcal{X}^{(t)} \times a^{(t)} \to \mathcal{X}^{(t+1)}$. Thus, any planning problem $\langle \mathcal{S}^{(init)}, \mathcal{S}^{(final)} \rangle$ can be formulated as a logical planning problem $\mathcal{Q} = \langle \mathcal{O}, \mathcal{D}, \mathcal{X}^{(init)}, \mathcal{X}^{(final)} \rangle$, which is solved by a symbolic planner to produce a task plan $a^{(0)}, a^{(1)}, \dots, a^{(T-1)} = \texttt{PDDLSolver}(\mathcal{Q})$. Each logical action $a$ must then be integrated with corresponding motion planning skills to generate continuous robotic actions $\tilde{a}$ for execution.

## 4 PROBLEM STATEMENT

Our system solves a new long-horizon planning problem given an expert demonstration trajectory $\tau_{demo}$ and its associated task description $T_{demo}$. The trajectory $\tau_{demo}$ is a sequence of continuous environment states $\mathcal{S}^{(0)}, \mathcal{S}^{(1)}, \dots$, while $T_{demo}$ is a short natural language phrase describing the task. Following prior work (Silver et al., 2023; Kumar et al., 2023; Huang et al., 2025), the robot is the sole acting agent during data collection.

Given a new planning problem, PDDLLM derives a PDDL domain from $(T_{demo}, \tau_{demo})$ which generates a sequence of symbolic actions as the task plan. LoCA, a component within PDDLLM, interfaces the task plan with an off-the-shelf motion planner, which produces executable robot trajectories. Formally, the problem can be expressed as:

$$\tilde{a}^{(0)}, \tilde{a}^{(1)}, \dots, \tilde{a}^{(L-1)} = MotionPlanner(PDDLLM(\mathcal{S}_{new}^{(init)}, \mathcal{S}_{new}^{(goal)}, T_{demo}, \tau_{demo})) \qquad (1)$$

where $\mathcal{S}_{new}^{(init)}$ and $\mathcal{S}_{new}^{(goal)}$ define the initial and goal states of the new problem, $L$ denotes the task plan length, and $\tilde{a}^{(0:L)}$ indicates an executable robot trajectory.

## 5 METHODOLOGY

Figure 1 presents an overview of the PDDLLM framework. With $T_{demo}$ and $\tau_{demo}$ as inputs, PDDLLM constructs a relevant predicate library through predicate imagination and generates an action library via action invention. Ultimately, these predicate and action libraries are compiled into an executable PDDL planning domain, automatically interfaced with motion planners via LoCA. In the following sections, we provide a detailed explanation of each step in the framework.

### 5.1 PDDL DOMAIN GENERATION

Given a human demonstration and its task description, our system combines simulated physical interactions with LLM-based reasoning to generate an executable PDDL planning domain through predicate imagination and action invention. Simulation verifies the physical feasibility that LLMs alone cannot reliably enforce, while the LLM abstracts these grounded interactions into logical predicates and actions. This combination enables a robust pipeline for planning-domain construction.

#### 5.1.1 PREDICATE IMAGINATION

Predicate imagination refers to the process of summarizing simulated object relation roll-outs into meaningful predicates. The process consists of two stages: Stage 1 generates first-order predicates and Stage 2 further derives the higher-order predicates from the first-order predicates.

**Stage 1.** In this stage, PDDLLM generates first-order predicates, which directly describe the physical properties or relations of the objects (e.g., $(\texttt{is\_on} \ ?o_1 \ ?o_2)$, $(\texttt{smaller} \ ?o_1 \ ?o_2)$), by summarizing simulated object interactions using an LLM.

**Definition 1** (Feature Space). *The feature space is defined as a set of continuous state variables, such as position, orientation, size, and color, that fully characterize the state of each object in the environment.*

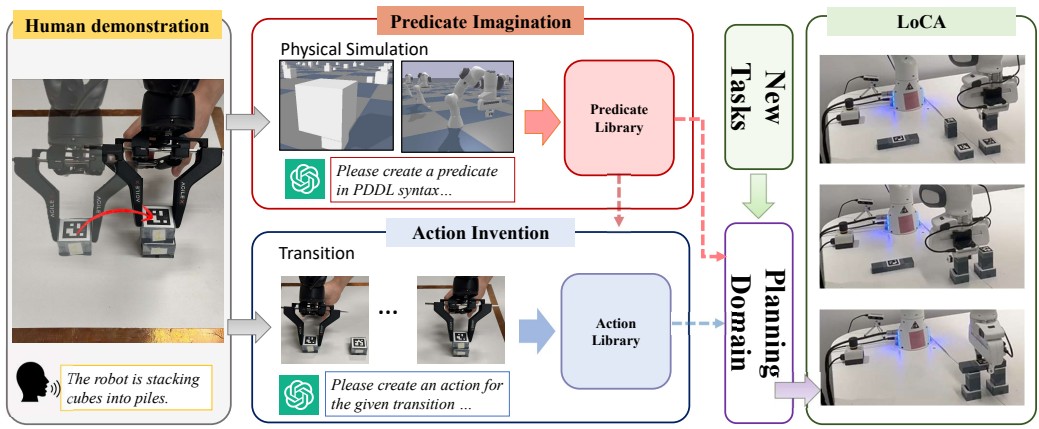

Figure 1: Overview of the proposed framework. (1) Human demonstrations, in the form of manipulation trajectories, and the corresponding task descriptions, serve as input. Implementation details is shown in Section B.12. (2) PDDLLM initiates thousands of parallel simulations, using the resulting roll-outs and rich physics-based feedback to guide the LLM in summarizing them into meaningful predicates, and returns a predicate library annotated with each predicate's relevance to the current task. (3) Actions are invented by an LLM that summarizes logical state transition patterns from the demonstration, which is grounded into logical states using the imagined predicates. (4) The predicates and actions are compiled into a planning domain, which is automatically interfaced with motion planning algorithm by the Logical Constrain Adapter (LoCA) to solve new tasks.

Following Definition 1, the feature space is defined as a set of variables, such as Cartesian coordinates $(x, y, z)$ for object positions and RGB values $(r, g, b)$ for colors. The feature space is bounded by real-world constraints. Object states are uniformly sampled across this space to span a diverse range of object-object and object-environment interactions. Each sampled state undergoes a two-step verification process. First, it is evaluated for physical feasibility using a physical simulator, which serves as a physical knowledge base to capture complex dynamics beyond the reasoning capacity of LLMs. Only physically valid states are retained.

However, the full continuous feature space is highly scattered, making it difficult for the LLM to detect consistent patterns or infer meaningful numeric thresholds. Thus, we discretize the feature space into coherent sub-regions that group similar states together, enabling the LLM to extract stable relational patterns. Additionally, these sub-feature-space boundaries provide physically grounded constraints that allow the motion planner to reliably evaluate predicate truth values during planning.

Next, PDDLLM partitions the feature space into a finite collection of subspaces. The range of each feature is divided into intervals, with the length of each interval being a hyperparameter. The intersections of these intervals specify the subspaces. Each object state can be mapped into one of the subspaces. Each subspace is analyzed to determine whether it contains feasible object states, as verified through simulation in the previous verification step. If so, simulation roll-out summaries are generated as prompts following a predefined scene-description template. Prompt generation is automated as it only requires the replacement of some keywords (such as "position" with "color") and the specification of interval boundaries. An LLM is then prompted to summarize subspaces into meaningful predicates and select those relevant to the task. The subspace boundaries serve as predicate physical constraints, enabling the classification of whether a predicate holds true. Figure 2.a illustrates an example of predicate generation for positional relations between objects.

**Stage 2.** Although the first-order predicates already capture all necessary features of the environment, higher-order predicates are essential for representing more complex relations and improving planning efficiency. Thus, we systematically derive higher-order predicates by combining first-order predicates with logical operators and quantifiers. We focus on using negation operator, together with the quantifiers "for all" ($\forall$) and "there exists" ($\exists$), as they are empirically proven effective in robotics by prior work (Curtis et al., 2022; Silver et al., 2023). These components are combined in all possible valid ways to form richer logical expressions that provide stronger guidance during planning. For example, (is_on ?$o_1$ ?$o_2$) can be negated to produce (not_is_on ?$o_1$ ?$o_2$). When combined with

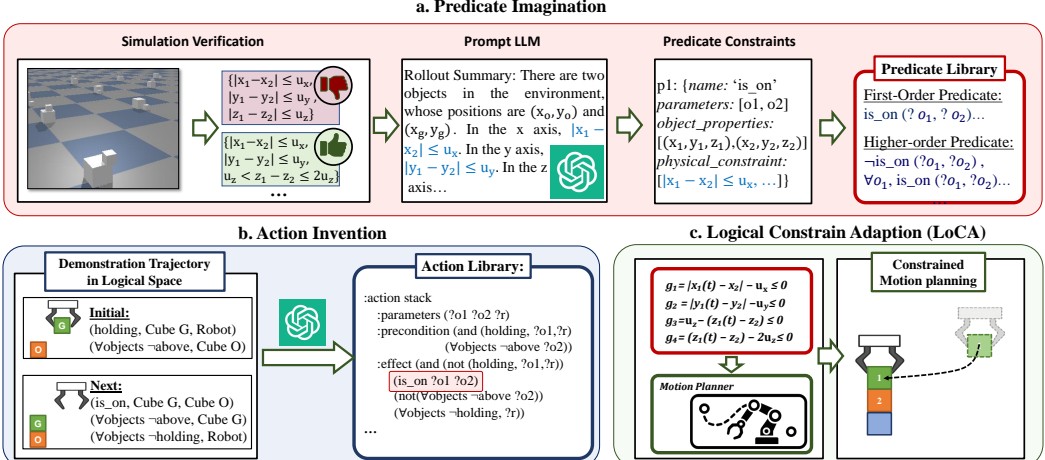

Figure 2: **a.** This example illustrates the imagination of predicates for relative object positions. Let $u$ be a configurable variable for each dimension. Object poses are sampled and simulated, with infeasible cases filtered out by the simulation feedback. Feasible subspaces are provided to the LLM to generate first-order predicates with their corresponding physical constraints. Higher-order predicates can be further derived using logical operators (e.g., "not", "for all") from first-order predicates. Diverse predicate examples are provided in Section B.9 **b.** This example shows how the *Stack* action is invented. Continuous states are grounded into logical states using the imagined predicates, where the state transition represents the logical action. By prompting the LLM with the pair of the current state and the next state after the transition, we obtain the PDDL definition of the action *Stack*. **c.** The integration of actions with the motion planner is handled automatically by LoCA, which retrieves the physical constraints associated with each first-order predicate in the action effect set $\mathcal{P}_{eff}$ and applies these constraints for motion planning.

the universal quantifier, this further yields ($\forall\_o_1\_not\_is\_on$ $?o_1$ $?o_2$), meaning that for any $o_1$ in the environment, ($is\_on$ $?o_1$ $?o_2$) is false. This indicates that object $o_2$ is on top.

### 5.1.2 ACTION INVENTION

After constructing the predicate library, the human demonstration $\tau_{demo}$ can be mapped into the logical space as $\tau_{demo}^{logic}$ by grounding all relevant predicates at each time step. The logical state transitions within $\tau_{demo}^{logic}$ signify the execution of actions. An advantage of learning actions in the logical space is that it simplifies pattern recognition by focusing on moments of logical state transitions, effectively transforming long trajectories into concise logical representations. For instance, the continuous manipulation trajectory $\tau_{demo}$ in Figure 2.b contains over 1000 time steps while $\tau_{demo}^{logic}$ is reduced to merely 2 steps. Logical state transitions are extracted from pre-and-post-change states and presented to the LLM as prompts. After experimenting with various prompt structures, we found this direct, structured method to be the most effective. A concrete example of action invention for *Stack* is elaborated in Figure 2.b.

### 5.1.3 PARALLEL PROMPTING WITH FEEDBACK

To avoid planning domain generation failure due to the random nature of LLMs, we adopt a parallel prompting strategy with domain execution feedback. Specifically, multiple candidate PDDL domains are generated from the same demonstration via prompting the LLM multiple times. If a generated domain encounters run-time error (syntax, incomplete domain, fail to reach goals...), then this planning domain is eliminated from the candidates. If more than one planning domain is successfully produced, then a majority vote would be initiated using the LLM to choose the best one. Empirically, we find that when the number of parallel prompts exceeds five, the correctness of the final domain converges to a stable level. Examples of the parallel prompting procedure are shown in Section B.7.

### 5.1.4 HYPERPARAMETER SELECTION

For each continuous feature $f$, we initialize its discretization scale $u_f$ using the smallest non-zero observed difference $d_{min}$ of that feature among all relevant objects. This provides a conservative upper bound on the required $u_f$: it ensures that the learned predicates remain expressive enough to capture distinctions for all objects. During LLM-based predicate summarization, the model is allowed to refine the initial discretization hyperparameter $u_f$ by either merging or subdividing sub-feature intervals. Specifically, the LLM can merge intervals that are semantically uninformative for the task into a single broader region, and subdivide intervals in feature regions where finer distinctions are necessary. Consequently, even if the initial choice of $u_f$ is imperfect, the refinement process enables the LLM to recover the task-relevant granularity and generate predicates that remain both expressive and compact.

### 5.2 AUTOMATIC INTERFACING WITH MOTION PLANNER

Combining the PDDL planning domain with symbolic solvers yields a task plan $a^{(0)}, a^{(1)}, \ldots, a^{(T-1)}$. The next step is to establish an interface between each logical action and the low-level motion planner.

In PDDLLM, this interfacing is fully automated through the Logical Constraint Adapter (LoCA). Prior approaches relied on manually encoding logical actions as mathematical constraints for motion planning (Toussaint, 2015). LoCA automates and generalizes this process by directly retrieving the physical constraints associated with the first-order predicates in an action's effect set $\mathcal{P}_{\text{eff}}$. These constraints, expressed as mathematical inequalities, are sequentially applied to the motion planner, ensuring that the required predicates are satisfied once the action is executed. In this way, LoCA automatically transforms each logical action into a standard constrained motion planning problem. As a result, the generated trajectory is guaranteed to align with the semantic meaning of the action, as illustrated in Figure 2.c. This eliminates the need for manually engineered interfaces or predefined motion-planning skills, thereby streamlining the integration between symbolic planning and low-level control.

Additionally, our framework is also compatible with recent advances such as vision–language–action (VLA) models by directly using logical actions as prompts. Experiments in Section 8 demonstrate PDDLLM's ability to automatically integrate with both traditional and learning-based motion planning.

## 6 EXPERIMENTS AND BASELINES

Building on evaluation practices in TAMP, we consider tasks with concrete conditions such as object position, orientation, size, and color (Silver et al., 2023; Garrett et al., 2020; Kumar et al., 2023; Liang et al., 2024; Huang et al., 2025). We structure our experiments along three dimensions, Task Diversity, Task Complexity, and Knowledge Transferability, whose definitions are provided in Section B.10.1, with task details described in Section B.4. Our experiments are conducted in PyBullet Simulation (Coumans & Bai, 2016–2021), with the symbolic solver from PDDLStream (Garrett et al., 2020) and the motion planning algorithm from PyBullet-Planning (Garrett et al., 2015; Garrett, 2018). We use GPT-4o (OpenAI, 2023), but our method is not tied to this backbone. All experiments use 10 parallel prompts and set $u_f = d_{min}$; full ablations are provided in Section B.15 and Section B.16.

By default, PDDLLM learns a planning domain from a single corresponding demonstration per task. To assess its ability to integrate cross-task knowledge, we also evaluated settings where only demonstrations from simpler related tasks were provided for a more complex one. Specifically, for the rearrangement, the model received only demonstrations of both stacking and unstacking, while for the bridge building, it was given only demonstrations of both stacking and alignment.

### 6.1 BASELINES AND ABLATIONS

We implement six baselines and one ablation of PDDLLM to comprehensively evaluate our method. GPT-4o is used as the default LLM unless otherwise specified, and the motion planning algorithm is kept the same for all methods. The complete details are provided in Section B.2

- **LLMTAMP**, **o1-TAMP**, **R1-TAMP**: LLM-based task and motion planning (LLMTAMP) inspired by Huang et al. (2022a); Li et al. (2022), which use LLMs for task planning, with language description of planning task as input. The task execution demonstration, same as those used in PDDLLM, was provided in the form of natural language in prompt. In addition to GPT-4o, reasoning LLMs, OpenAI's o1 (OpenAI, 2024) and Deepseek's R1 (DeepSeek-AI, 2025), are used as backbones for o1-TAMP and R1-TAMP, respectively.
- **LLMTAMP-FF**: Following the method by Huang et al. (2022b); Chen et al. (2024a), LLMTAMP-FF extends LLMTAMP with a failure feedback loop.
- **LLMTAMP-FR**: Following Wang et al. (2024), LLMTAMP-FR extends failure detection by providing specific failure reasons to guide replanning with the LLM.
- **Expert Design**: The expert design baseline uses expert-crafted planning domains with symbolic solvers. Expert-designed domains are refined from PDDLLM-derived domains by an expert.

In addition to the six baselines, we include an ablation of our method, **RuleAsMem**, which uses the generated PDDL domain as contextual memory for LLM task planners, replacing symbolic solvers.

Robot planning is required to be real-time in robot deployment, imposing constraints on the planning time allowance (Gammell et al., 2015; Garrett et al., 2015; 2020). In our experiment, a uniform planning time limit of 50 seconds is applied to all planning problems and methods. We measure performance using the planning success rate, as in other studies (Huang et al., 2025; Silver et al., 2023; Kumar et al., 2023). The planning time and token cost are used to measure the planning cost (Zhong et al., 2024). Three parallel runs were conducted to compute the mean and standard error for the planning success rate.

Table 1: Planning success rate (%) across tasks for all methods (time limit = 50 s). The best results are highlighted in bold. `Expert` is excluded from the comparison, as it requires additional manual effort and serves as an upper bound.

| Method | Expert | LLMTAMP | LLMTAMP-FF | LLMTAMP-FR | RuleAsMem | PDDLLM |
|---|---|---|---|---|---|---|
| Stack | $98.5 \pm 0.8$ | $41.7 \pm 4.3$ | $70.8 \pm 1.4$ | $64.2 \pm 3.1$ | $85.5 \pm 2.9$ | $\textbf{97.5} \pm \textbf{1.6}$ |
| Unstack | $100 \pm 0.0$ | $89.4 \pm 1.5$ | $94.6 \pm 0.9$ | $92.1 \pm 2.3$ | $88.4 \pm 1.2$ | $\textbf{97.7} \pm \textbf{0.7}$ |
| Color Classification | $100 \pm 0.0$ | $18.1 \pm 1.5$ | $36.4 \pm 1.1$ | $49.0 \pm 3.0$ | $88.7 \pm 2.3$ | $\textbf{100} \pm \textbf{0.0}$ |
| Alignment | $100 \pm 0.0$ | $31.1 \pm 3.1$ | $52.0 \pm 2.7$ | $40.0 \pm 2.4$ | $96.0 \pm 0.8$ | $\textbf{100} \pm \textbf{0.0}$ |
| Parts Assembly | $98.9 \pm 0.6$ | $33.3 \pm 1.5$ | $53.9 \pm 1.1$ | $41.3 \pm 1.2$ | $95.0 \pm 0.6$ | $\textbf{100} \pm \textbf{0.0}$ |
| Rearrange | $73.3 \pm 0.6$ | $5.6 \pm 1.0$ | $17.4 \pm 1.1$ | $11.8 \pm 1.8$ | $1.1 \pm 0.6$ | $\textbf{64.3} \pm \textbf{0.7}$ |
| Burger Cooking | $100 \pm 0.0$ | $27.8 \pm 2.8$ | $50.0 \pm 4.8$ | $48.6 \pm 6.9$ | $27.8 \pm 2.8$ | $\textbf{91.7} \pm \textbf{4.8}$ |
| Bridge Building | $100 \pm 0.0$ | $43.3 \pm 3.3$ | $53.3 \pm 3.8$ | $51.7 \pm 2.5$ | $20.0 \pm 0.0$ | $\textbf{87.2} \pm \textbf{4.3}$ |
| Tower of Hanoi | $100 \pm 0.0$ | $14.3 \pm 0.0$ | $14.3 \pm 0.0$ | $14.3 \pm 0.0$ | $14.3 \pm 0.0$ | $\textbf{100} \pm \textbf{0.0}$ |
| **Overall** | $95.7 \pm 0.1$ | $35.7 \pm 0.5$ | $52.5 \pm 0.4$ | $48.6 \pm 0.8$ | $69.9 \pm 0.7$ | $\textbf{93.3} \pm \textbf{0.7}$ |

## 7 RESULTS

Through the experiments, we aim to answer the following research questions: (1) How does PDDLLM perform relative to other LLM-based planners? (2) Can PDDLLM generalize to unseen, more complex tasks? (3) Does PDDLLM derive high-quality domains with performance comparable to expert designs? (4) How does PDDLLM's token cost compare to other LLM-based planners? (5) How does PDDLLM recover from errors such as by suboptimal demonstrations?

**Q1. Performance comparison to baselines:** Table 1 presents the planning success rate of all evaluated methods across all tasks, measured with a 50-second time limit. PDDLLM shows clear advantages in planning efficiency and generalizability over baseline methods. While LLM-based baselines perform competitively in simpler tasks like stacking and unstacking, their performance drops sharply in complex tasks such as rearrangement, burger cooking, bridge building, and Tower of Hanoi. In contrast, PDDLLM maintains strong performance across all task categories, achieving an over 40% improvement in overall planning success rate compared to the best LLM-based planner baseline (i.e., LLMTAMP-FF). Even the ablated variant of PDDLLM, RuleAsMem, outperforms the LLM-based

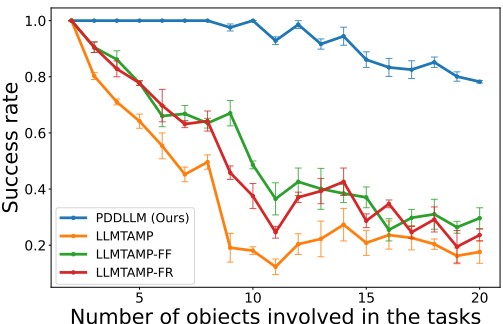 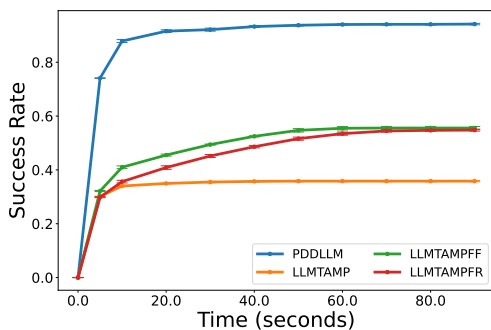

Figure 3: (left) Planning success rate trend across increasing object counts. (right) Overall planning success rate under varying time limits.

planners by at least 17.4% in overall success rate. Compared to PDDLLM, RuleAsMem exhibits less stability. It performs well in simpler tasks but struggles in more complex ones, suggesting the LLMs struggle to understand complex domains. In addition to the results evaluated with a 50-second time window, we also report the overall success rates of the main methods across varying time limits. As shown in Figure 3(right), PDDLLM consistently outperforms the baseline methods and demonstrates superior planning performance across all time limits. Notably, it reaches performance saturation the fastest, highlighting its superior time efficiency among all evaluated approaches.

We further compare PDDLLM's planning ability with more powerful reasoning LLMs (OpenAI-o1 and DeepSeek-R1) in Table 2. While reasoning-based models exhibit strong planning capabilities, their high computational cost prevents them from completing within the 50-second window. To sufficiently compare the planning capabilities, we extend the time limit to 500 seconds and evaluate them on the most challenging tasks. Although o1-TAMP and R1-TAMP show remarkable improvement compared to GPT-4o-based LLMTAMP, they remain less robust in long-horizon planning and fail to match PDDLLM's performance. In contrast, our method, relying solely on GPT-4o, consistently achieves higher success rates across complex tasks. Detailed time costs for reasoning models are provided in Section B.6.

**Q2. Generalization:** The experimental results highlight PDDLLM's ability to handle increasing complexity in both planning and domain derivation, consistently outperforming baseline methods. In terms of planning complexity, as shown in Figure 3(left), PDDLLM maintains robust planning performance as task complexity grows, achieving high success rates even in scenarios involving up to 20 objects. As shown in Table 1, performance degradation is observed in more challenging tasks such as rearrangement, where longer action sequences are required and there are more complex motion constraints. From the perspective of domain derivation complexity, PDDLLM remains effective even in tasks demanding the generation of over 100 predicates. However, success rates drop in the most complex domains, such as bridge building, primarily due to missing supporting predicates. More details of the generalization of each task are provided in section B.5.

PDDLLM demonstrates a strong ability to integrate knowledge across demonstrations. The modular nature of PDDL action syntax allows the easy transfer of actions learned from different demonstrations among domains. Despite receiving only one example each for stacking and unstacking, it successfully combines the "stack" action and the "unstack" action to solve rearrangement tasks. Similarly, in the bridge building domain, PDDLLM successfully combines the "align" action and the "stack" action using one demonstration of cube alignment and one of stacking. The high success rate of these tasks in Table 1 underscores the robustness of the derived planning domains.

**Q3. Domain Quality:** We evaluate the quality of planning domains generated by PDDLLM by comparing the percentage of missing or redundant predicates(actions) and the planning success rate against expert-designed domains. The full definition of missing or redundant can be found in Section B.11. None of the generated domains are missing actions. Table 3 reports the percentage of missing and redundant predicates for the four evaluated tasks, measured against human-designed domains. The number of parallel prompts is fixed at 10, and each task is evaluated over three runs; the reported values are the averages across these runs. These results show that PDDLLM produces high-quality domains with minimal errors, even in complex scenarios. While a few predicates may be

absent, the overall logical structure remains sound, as reflected in the high overall planning success rate of 93.3% shown in Table 1, closely matching the performance of expert-crafted domains.

**Q4. Token efficiency:** We compare the token efficiency of our method against LLM-based baselines on the three most complex tasks, as shown in Table 2. Although the GPT-4o-based LLMTAMP incurs lower total token costs, it performs poorly across all three tasks. Compared to O1-TAMP and R1-TAMP, PDDLLM uses significantly fewer tokens while consistently achieving better performance. These results also underscore the challenge of deploying reasoning LLMs on real robot systems for long-term use due to high token consumption. Unlike other LLM-based planners, PDDLLM spends tokens only during domain derivation; execution is handled by a PDDL solver with no additional token cost. This makes it well suited for long-term deployments that repeatedly perform similar tasks.

Table 2: Comparison of planning success rate (%) and token cost (k) between PDDLLM and LLMTAMP and the reasoning LLM variants. The best results are shown in bold, and the second-best results are underlined.

| Task | Success Rate (%) ↑ | | | | Token Cost (k) ↓ | | | |
|---|---|---|---|---|---|---|---|---|
| | **PDDLLM** | **LLMTAMP** | **O1-TAMP** | **R1-TAMP** | **PDDLLM** | **LLMTAMP** | **O1-TAMP** | **R1-TAMP** |
| Rearrangement | **73.8 ± 1.1** | 5.6 ± 1.0 | 70.8 ± 1.5 | 40.0 ± 5.0 | 334 | **212** | 1200 | 1460 |
| Tower of Hanoi | **100.0 ± 0.0** | 14.3 ± 0.0 | 33.3 ± 2.4 | 14.3 ± 0.0 | 535 | **36** | 529 | 353 |
| Bridge Building | **87.2 ± 4.3** | 44.3 ± 3.3 | 51.7 ± 2.5 | 40.0 ± 0.0 | 375 | **50** | 270 | 363 |
| Overall | **80.5 ± 0.5** | 13.9 ± 0.9 | 61.5 ± 1.3 | 35.9 ± 3.1 | 415 | **99** | 666 | 725 |

**Q5. Domain Correction:** Domain refinement can occur by adding demonstrations or refining language guidance. To evaluate how PDDLLM corrects an imprecise domain, we conducted bridge-building experiments that require coordinated picking, stacking, and alignment skills, using demonstrations that were intentionally masked or perturbed. With only a picking demo, PDDLLM had no evidence that stacking or alignment mattered, so their predicates were overlooked or collapsed into vague forms. As additional demonstrations were provided, PDDLLM progressively reconstructed the missing structure and sharpened constraints. Table 4 shows success rates rising as the missing information is supplied, demonstrating effective domain repair. We also injected a noisy, unrelated unstacking demo; it briefly appeared in the domain but was later pruned due to low usage frequency.

In addition to supplying more demonstrations, providing clearer task descriptions also improves predicate precision by offering richer contextual cues about the task objective. We include an experiment validating this effect in Section B.18.

Table 3: Percentage of missing or redundant predicates and actions across tasks.

| Task | Missing | Redundant |
|---|---|---|
| Stack | 4.2% | 8.3% |
| Burger Cooking | 22.2% | 3.7% |
| Bridge Building | 22.2% | 3.7% |
| Tower of Hanoi | 0.0% | 14.3% |

Table 4: Bridge-building success rate (%) under varying demonstration conditions.

| Class | Demonstration Type | Success Rate (%) ↑ |
|---|---|---|
| Missing | pick | 0.0 ± 0.0 |
| Missing | pick & stack | 20.0 ± 0.0 |
| Complete | pick & stack & align | 86.7 ± 3.8 |
| Redundant | pick & stack & align & unstack | 83.3 ± 3.3 |

# 8 REAL ROBOT DEPLOYMENT

We validate PDDLLM on real robots to demonstrate cross-platform deployability. The system is tested on the Agilex Piper, Franka Panda, and the UR5e, using ArUco markers for pose estimation and ROS2 for control. All platforms successfully complete tasks, including table-top stacking, bridge building, burger cooking, and the Tower of Hannoi through direct deployment of PDDLLM (details in Section B.1). We also evaluated how PDDLLM integrates with learned skills such as VLA on a tabletop stacking task with a UR5e. In this experiment, logical actions serve directly as prompts to condition policies,, bypassing explicit motion constraints. The real-robot experiment with VLA (details in Section B.1.3) demonstrates PDDLLM's compatibility with both classical and learning-based control.

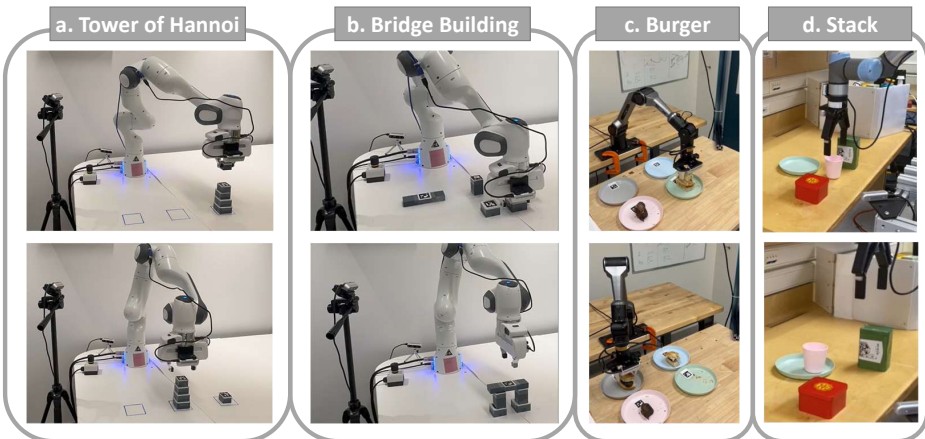

Figure 4: Real-robot experiment in three different platforms

# 9 LIMITATION

A limitation of PDDLLM is its occasional omission of highly complex predicates, which can reduce problem-solving efficiency and lead to lower success rates in more difficult tasks such as Burger Cooking and Bridge Building (Table 1). For example, in bridge building, the system may attempt to assemble the surface before the base is complete, making subsequent placements infeasible and requiring backtracking. Expert-designed domains address this by introducing predicate `(all_base_finished)` to enforce correct ordering. While the absence of such predicates does not directly cause planning failure, it slows down planning as the planner must perform additional feasibility checks and degrade performance under a fixed time budget. Recent work has attempted to mitigate this issue by refining planning domains through interactive or environment-feedback-based methods (Liang et al., 2024; Huang et al., 2025; Zhu et al., 2025). As with many TAMP systems, perception errors and the representation of complex dynamics and geometries, such as deformable objects and fluids, remain challenging for PDDLLM. Thus, we demonstrate that PDDLLM can be combined with approaches designed for such settings, including VLA-based manipulation methods.

# 10 FUTURE WORKS

There are a few future work directions we found meaningful to our approach. (1) Integrating perception to enable domain derivation from raw sensory inputs. Symbolic planning struggles to ground raw visual inputs into logical representations. Learning visual features directly would allow the system to derive predicates from sensor data, improving robustness to perception noise and reducing reliance on perception engineering. (2) Enabling direct interaction with the environment allows the system to obtain feedback that helps complete missing predicates, improve domain quality, reduce reliance on simulation, and better capture complex real-world dynamics. (3) Partial observability environments that requires active explore to acquire missing information.

# 11 CONCLUSION

This paper presents PDDLLM, the first approach in the field to generate a complete planning domain from scratch, without relying on any predefined predicates or actions. By extracting logical structures directly from pre-trained LLMs, PDDLLM autonomously derives both predicates and actions, enabling fully automated domain construction. Evaluated across a wide range of environments, PDDLLM demonstrates high quality in domain derivation and strong generalizability across diverse task categories. Moreover, when integrated with the LoCA framework, PDDLLM fully automates the integration between the PDDL planning domain and the low-level motion planner. This level of automation significantly improves usability and positions the framework as an adaptable and scalable solution for robotic planning and decision-making. Compared to existing methods, PDDLLM outperforms other LLM-based baselines and closely matches the performance of expert-designed planning domains, particularly in complex and long-horizon planning scenarios.

REPRODUCIBILITY STATEMENT

To facilitate reproduction of our results, we provide clear references to the relevant sections of the paper. The overall planning pipeline is described in Section 4. The extraction of the PDDL domain from demonstrations is illustrated in Figure 1, with additional details provided in Section B.12. The procedure for predicate imagination and action invention is discussed in the Section 5, where we also outline the prompt templates and prompting strategies in greater detail in Section B.3. Finally, the LoCA framework is described comprehensively in Section 5.2, and experiment details shown in Section 8 and Section B.1.

ETHICS STATEMENT

We affirm our commitment to the principles laid out in the ICLR Code of Ethics, including honesty, fairness, transparency, and avoidance of harm. The human demonstrations collected do not contain identifying information. Our research does not use sensitive personal or private data. We have carefully documented our methodology, data sources, and limitations to promote reproducibility and accountability. We see no foreseeable misuses of the methods or results discussed in this work, but we remain alert to potential downstream impacts and encourage continued scrutiny in the broader research community.

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

# A   THE USE OF LARGE LANGUAGE MODELS (LLMs)

This paper evaluates LLMs within the robotics domain, where LLMs serve as the testing target of our method. Beyond this role, LLMs were used as grammar-checking and text-polishing tools to improve readability (e.g., grammar, clarity, and style). They were not involved in the ideation, experimental design, analysis, or interpretation of results, and did not contribute to the scientific content. The authors take full responsibility for the entirety of the work presented.

# B   APPENDIX

## B.1   REAL ROBOT EXPERIMENT

Ten runs of each real robot experiment shown in Section 8 are performed. The success rates of real-world deployment on different robot platforms are shown as below:

Table 5: Real-world success rate across tasks.

| Metric | Tower of Hanoi (Franka) | Bridge Building (Franka) | Burger (Piper) | Table-top Stacking (UR5e) |
|---|---|---|---|---|
| Success Rate | 9/10 | 8/10 | 7/10 | 7/10 |

### B.1.1   FRANKA PANDA ARM

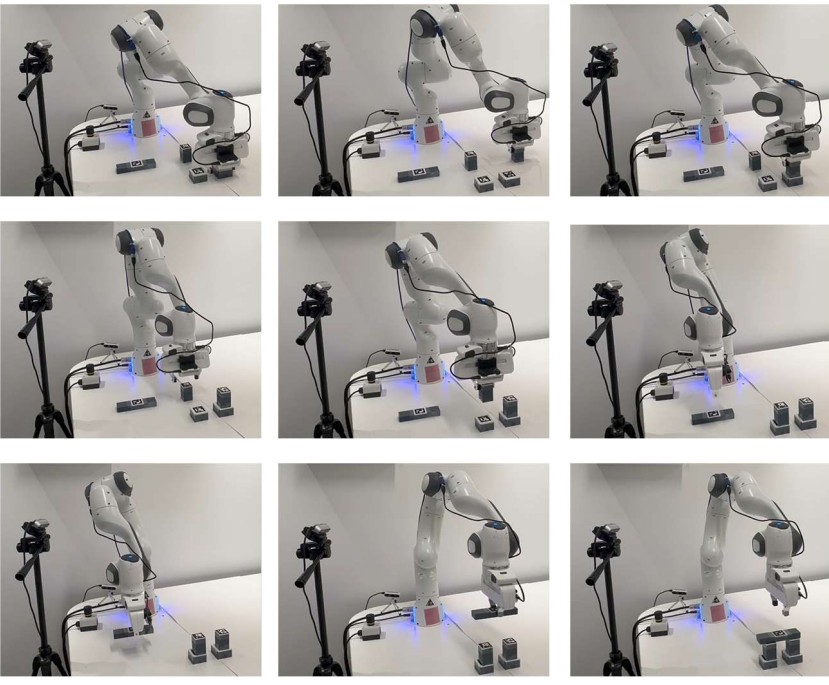

Figure 5: Franka Panda Arm building a bridge

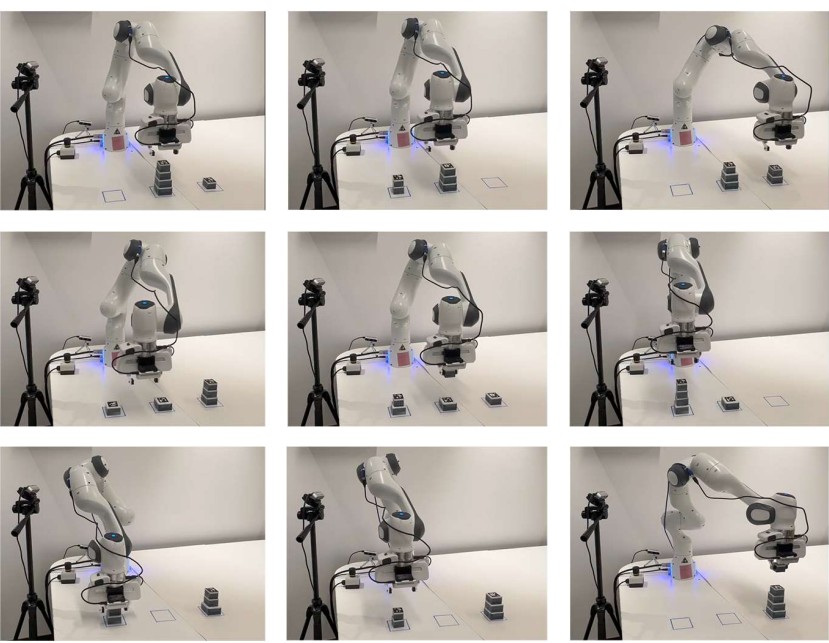

Figure 6: Franka Panda Arm solving the Tower of Hannoi puzzle

### B.1.2 AGILEX PIPER ARM

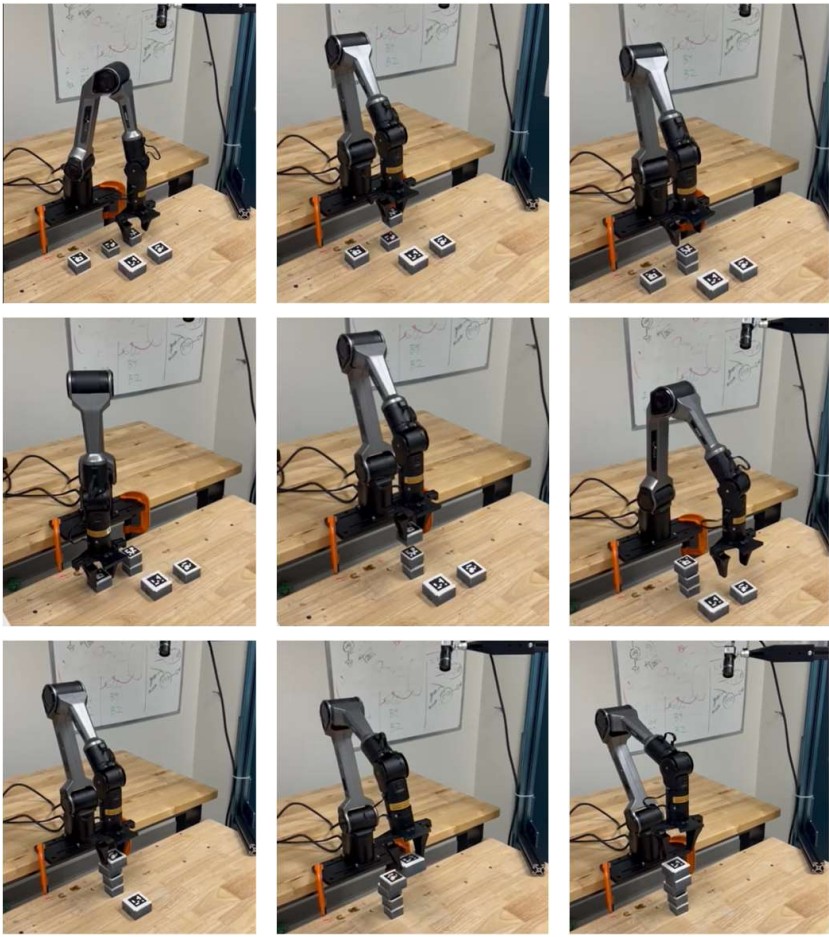

Figure 7: Agilex Piper Arm stacking cubes

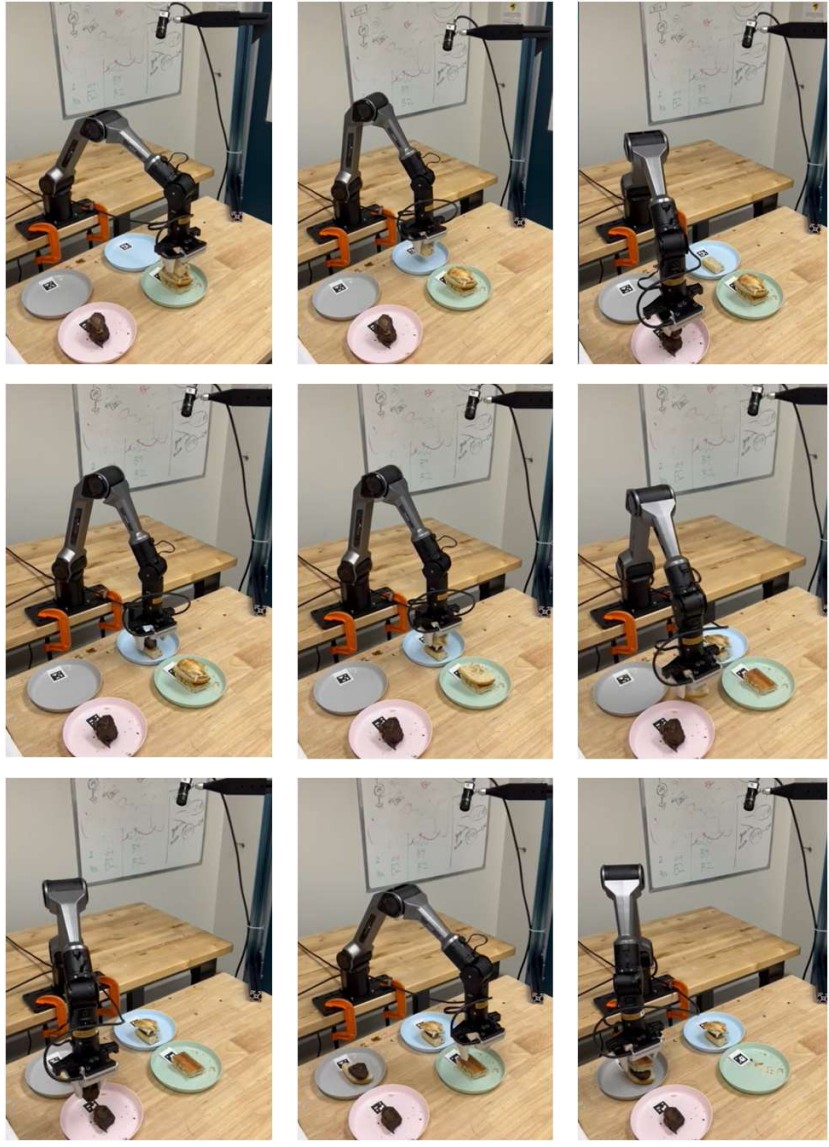

Figure 8: Agilex Piper Arm making burgers

### B.1.3 VLA Experiment with UR5e

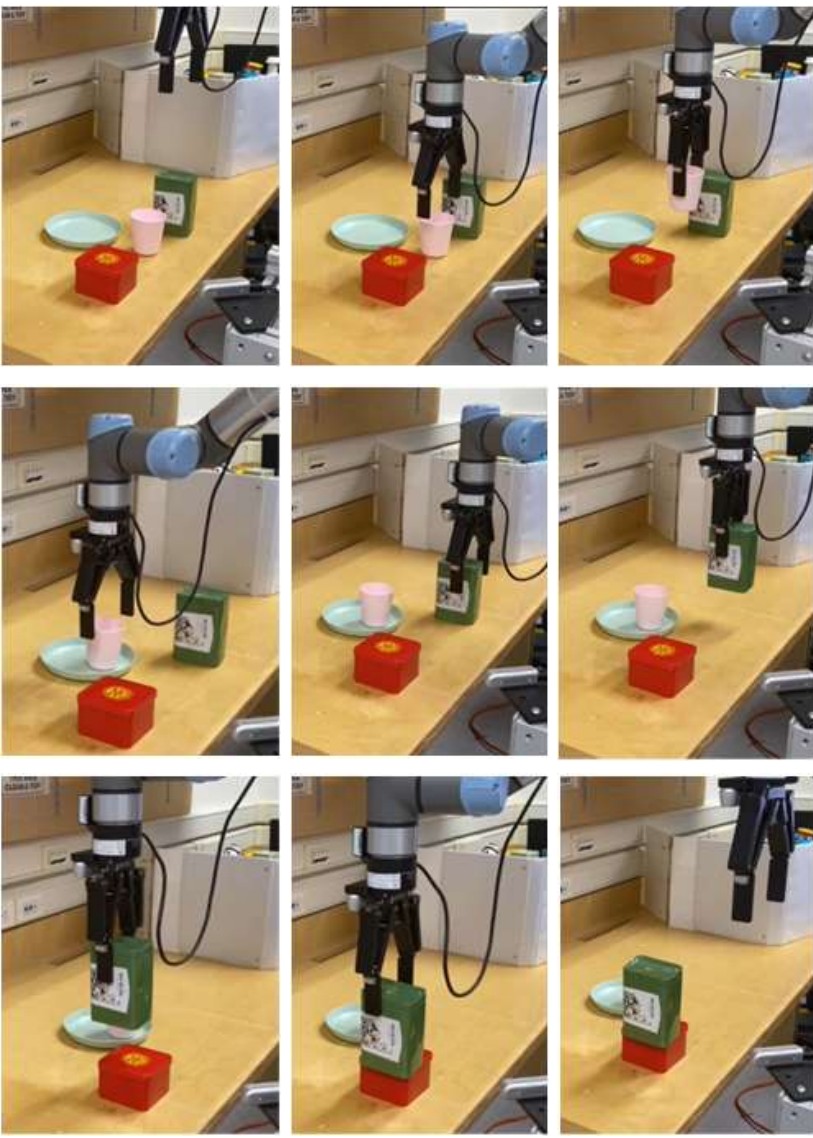

Figure 9: UR5e Arm Stacking Tabletop Objects

Experiment with the vision-language-action model is conducted using a $\pi 0$ model (Black et al., 2024), finetuned on 200 real-world pick and place trajectories. The resulting policy serves as the action module within our pipeline and is invoked by the PDDLLM planning system. In this setup, PDDLLM generates a high-level task plan, while the execution of each logical action in the plan is carried out directly by the Pi0 policy. Importantly, the VLA experiment demonstrates the flexibility of our framework: the action execution layer can be implemented with either a classical motion planner or a learning-based policy, depending on the task requirements.

## B.2 BASELINE IMPLEMENTATION DETAILS

**LLMTAMP**: LLM-based task and motion planning (LLMTAMP) builds on methods from (Huang et al., 2022a; Li et al., 2022), which use pre-trained LLMs for task planning. We formulate the planning problem as a natural language description. Human demonstrations are interpreted as action sequences required to accomplish the task. The LLM then generates high-level actions to achieve the goal, which are refined into motion plans using predefined skills.

**LLMTAMP-FF**: LLMTAMP-FF, following the method by (Huang et al., 2022b; Chen et al., 2024a), extends LLMTAMP with a failure feedback loop. Upon execution failure, the system feeds the failure signal to the LLM to regenerate the task plan, repeating until success or the time limit is reached.

**LLMTAMP-FR**: Following Wang et al. (2024), LLMTAMP-FR extends failure detection by providing specific failure reasons to guide replanning with the LLM. We design a reasoner that generates detailed explanations for plan failures and incorporates them into the prompt as feedback. The LLM performs failure reasoning and then regenerates the plan accordingly.

**Expert Design**: The expert design baseline uses expert-crafted planning domains to evaluate how closely PDDLLM-generated domains approach human-level performance. To highlight the readability and customizability of PDDLLM, these expert-designed domains are initialized with PDDLLM-generated outputs, which are then analyzed and refined by a TAMP expert into ground-truth domains.

**o1-TAMP and R1-TAMP**: As reasoning models have demonstrated superior performance in many tasks (Zhong et al., 2024), we ablate the LLM in LLMTAMP to compare the planning performance of state-of-the-art reasoning LLMs with our method. Specifically, we evaluate OpenAI's o1 (OpenAI, 2024) and Deepseek's R1 (DeepSeek-AI, 2025) as the reasoning backbones.

**RuleAsMem**: RuleAsMem is an ablation of PDDLLM that treats the generated PDDL domain as contextual memory, rather than using it with a symbolic planner. While prior work focuses on translating language into logical representations (Liu et al., 2023; Xie et al., 2023), RuleAsMem directly integrates logical planning rules into the LLM prompt to solve new tasks. Each task is defined by initial and goal states, using the imagined predicates, along with a human demonstration in the form of a PDDL task plan as prompts.

### B.3 PROMPT TEMPLATE

### B.3.1 PREDICATE IMAGINATION

**Template**

*There are n objects in the environment, whose feature name are feature value 1 and feature value 2. In dimension 1, we know feature subspace range in dimension 1. In dimension 2, we know feature subspace range in dimension 2.Please create a predicate in PDDL syntax to describe this relation and classify if it is related to the current task description. Please return the result in the following format: predicate, relevance.*

---

**Example of object position relation**

Prompt: There are two objects in the environment, whose positions are $(x_1, y_1)$ and $(x_2, y_2)$. In the horizontal direction, $|x_1 - x_2| \leq u$. In the vertical direction, $u < y_1 - y_2 \leq 2u$. Please create a predicate in PDDL syntax to describe this relation and classify if it is related to the task of stacking cubes together. Please return the result in the following format: predicate, relevance.

LLM: is_on($? o_1$ , $? o_2$ ), related.

---

**Example of color relation**

Prompt: There are two objects in the environment, whose colors are $(r_1, g_1, b_1)$ and $(r_2, g_2, b_2)$. In the red channel, $|r_1 - r_2| \leq u$. In the green channel, $|g_1 - g_2| \leq u$. In the blue channel, $|b_1 - b_2| \leq u$. Please create a predicate in PDDL syntax to describe this relation and classify if it is related to the task of stacking cubes together. Please return the result in the following format: predicate, relevance.

LLM: similar_color($? o_1$ , $? o_2$ ), unrelated.

---

**Example of size relation**

Prompt: There are two objects in the environment, whose size are $(w_1, l_1, h_1)$ and $(w_2, l_2, h_2)$. In the width, $0 < w_1 - w_2 \leq u$. In the length, $0 < l_1 - l_2 \leq u$. In the height, $0 < h_1 - h_2 \leq u$. Please create a predicate in PDDL syntax to describe this relation and classify if it is related to the task of stacking cubes together. Please return the result in the following format: predicate, relevance.

LLM: larger_than($? o_1$ , $? o_2$ ), related.

---

*Note: Here, $u$ is a variable determined by feature subspace range length, which depends on the user input.*

### B.3.2 ACTION INVENTION

**Template**

*Current state: The logical state before action execution*

*Next state:The logical state after the action execution*

*Can you give this transition an action name to summarize and describe what happened. Please follow Planning Domain Definition Language syntax.*

---

**Example for inventing 'Pick'**

Prompt: The current state of the environment is:
(∀objects ¬above, o1), (is_on, o1 , o2), (∀objects ¬holding, robot)
After one action execution, the resulting state of the environment is:
(holding, o1, robot), (∀objects ¬above, o2)
Can you name this logical action to summarize and describe what happened? Please follow Planning Domain Definition Language syntax and return together with action parameters, preconditions, and effects.
LLM:
:action pick
    :parameters (?o1 ?o2 ?r)
    :precondition (and (is_on ?o1 ?o2) (∀objects ¬above ?o1) (∀objects ¬holding, ?r))
    :effect (and (not (is_on ?o1 ?o2)) (not (∀objects ¬holding, ?r) (∀objects ¬above, ?o2)
        (holding, ?o1,?r) (not (∀objects ¬above, o1)))

---

**Example for inventing 'Place'**

Prompt: The current state of the environment is:
(holding, o1, robot) (∀objects ¬above, o2)
After one action execution, the resulting state of the environment is:
(∀objects ¬above, o1) (∀objects ¬above, o2) (∀objects ¬holding, robot)
Can you name this logical action to summarize and describe what happened? Please follow Planning Domain Definition Language syntax and return together with action parameters, preconditions, and effects.
LLM:
:action place
    :parameters (?o1 ?r)
    :precondition (and (holding, ?o1,?r))
    :effect (and (not (holding, ?o1,?r)) (∀objects ¬above ?o1) (∀objects ¬holding, ?r))

### B.3.3 LLMTAMP BASELINE

**Part 1. Initialization:**

*Imagine you are a robot arm operator; you need to generate a sequence of actions to achieve the given goal. Here are the logical actions you can choose from: list of actions to choose from*

**Part 2. Human demonstrations:**

*Here are some examples for you to learn:*

*example of input and output of the system*

**Part 3. New planning problem:**

*Now you are given a new input planning problem as the following: The initial state and the goal of the problem to be solved Choose a sequence of actions to accomplish this task, and return the action sequence following the example output provided.*

We integrate parts 1, 2, and 3 into a complete prompt for the LLM to generate task plans.

---

**Example for a stacking problem**

Prompt: Imagine you are a robot arm operator; you need to generate a sequence of actions to achieve the given goal. Here are the logical actions you can choose from: stack(upper box, lower box, robot), pick(upper box, table, robot).

Here are some examples for you to learn:
Example input: Object 0 is a robot. Object 1 is a table. Object 2 is a box. Object 3 is a box. Object 4 is a box. Initially, the robot is not holding anything. Object 2 is on table. Object 3 is on table. Object 4 is on table. Object 2 is the topmost object. Object 3 is the topmost object. Object 4 is the topmost object. In the goal, Object 2 is above Object 3. Object 3 is above object 4. Object 4 is on Object 1. Object 2 is the topmost object. The robot is not holding anything.
Example Output: pick(3, 1, 0), stack(3, 4, 0), pick(2, 1, 0), stack(2, 3, 0).

Now you are given a new planning problem as the following: Object 0 is a robot. Object 1 is a table. Object 2 is a box. Object 3 is a box. Object 4 is a box. Initially, Object 2 is on the table. Object 3 is on the table. Object 4 is on the table. Object 2 is the topmost object. Object 3 is the topmost object. Object 4 is the topmost object. The robot is not holding anything. In the goal, Object 2 is above object 3. Object 4 is above object 2. Object 3 is on Object 1. Object 4 is the topmost object. The robot is not holding anything. Choose a sequence of actions to accomplish this task, and return the action sequence following the example output provided.

LLM: pick(2, 1, 0), stack(2, 3, 0), pick(4, 1, 0), stack(4, 2, 0).

---

### B.3.4 LLMTAMP+FAILURE FEEDBACK BASELINE

LLMTAMP+Failure Feedback extends LLMTAMP with a failure feedback loop. Upon execution failure, the system feeds the failure signal to the LLM for replanning. Thus, the initial prompts of this baseline, part 1, 2, and 3, are the same as the LLMTAMP prompt. However, there is a failure summarization. Integrating part 1-4 gives the full prompt.

**Part 4. Failure feedback:**

*Your plan failed in execution, please generate a different one. Only return the sequence of logical actions following the format of example output.*

---

**Example for a stacking problem**

Prompt: Imagine you are a robot arm operator; you need to generate a sequence of actions to achieve the given goal. Here are the logical actions you can choose from: stack(upper box, lower box, robot), pick(upper box, table, robot).

Here are some examples for you to learn:
Example input: Object 0 is a robot. Object 1 is a table. Object 2 is a box. Object 3 is a box. Object 4 is a box. Initially, the robot is not holding anything. Object 2 is on table. Object 3 is on table. Object 4 is on table. Object 2 is the topmost object. Object 3 is the topmost object. Object 4 is the topmost object. In the goal, Object 2 is above Object 3. Object 3 is above object 4. Object 4 is on Object 1. Object 2 is the topmost object. The robot is not holding anything.
Example Output: pick(3, 1, 0), stack(3, 4, 0), pick(2, 1, 0), stack(2, 3, 0).

Now you are given a new planning problem as the following: Object 0 is a robot. Object 1 is a table. Object 2 is a box. Object 3 is a box. Object 4 is a box. Initially, Object 2 is on the table. Object 3 is on the table. Object 4 is on the table. Object 2 is the topmost object. Object 3 is the topmost object. Object 4 is the topmost object. The robot is not holding anything. In the goal, Object 2 is above object 3. Object 4 is above object 2. Object 3 is on Object 1. Object 4 is the topmost object. The robot is not holding anything. Choose a sequence of actions to accomplish this task, and return the action sequence following the example output provided.

LLM: pick(4, 1, 0), stack(4, 2, 0), pick(2, 1, 0), stack(2, 3, 0).

Prompt: Your plan failed in execution, please generate a different one. Only return the sequence of logical actions following the format of example output.

LLM: pick(2, 1, 0), stack(2, 3, 0), pick(4, 1, 0), stack(4, 2, 0).

---

### B.3.5 LLMTAMP+FAILURE REASONING BASELINE

LLMTAMP + Failure Reasoning further extends failure detection by providing specific failure reasons to guide replanning with the LLM. Part 1, 2, and 3 are the same as the LLMTAMP prompt. Integrating part 1-5 gives the full prompt.

**Part 4. Failure Reasoning:**

*Your plan failed in execution, please generate a different one. This may involve sample new plans or reorder the last plan. Please generate output step-by-step, which includes your reasoning for the failure of the last plan. Answer the questions: (i) what is the cause of the failure of last plan? (ii) do you see similar mistakes in other steps in the plan? Here are the failure reasons: failure reasons*

**Part 5. Replan:**

*Now, based on your above failure reasoning for (i) and (ii) , generate a new task plan to solve the original planning problem. Only return the sequence of logical actions following the format of example output.*

---

**Example for a stacking problem**

Prompt: Imagine you are a robot arm operator; you need to generate a sequence of actions to achieve the given goal. Here are the logical actions you can choose from: stack(upper box, lower box, robot), pick(upper box, table, robot).

Here are some examples for you to learn:
Example input: Object 0 is a robot. Object 1 is a table. Object 2 is a box. Object 3 is a box. Object 4 is a box. Initially, the robot is not holding anything. Object 2 is on table. Object 3 is on table. Object 4 is on table. Object 2 is the topmost object. Object 3 is the topmost object. Object 4 is the topmost object. In the goal, Object 2 is above Object 3. Object 3 is above object 4. Object 4 is on Object 1. Object 2 is the topmost object. The robot is not holding anything.
Example Output: pick(3, 1, 0), stack(3, 4, 0), pick(2, 1, 0), stack(2, 3, 0).

Now you are given a new planning problem as the following: Object 0 is a robot. Object 1 is a table. Object 2 is a box. Object 3 is a box. Object 4 is a box. Initially, Object 2 is on the table. Object 3 is on the table. Object 4 is on the table. Object 2 is the topmost object. Object 3 is the topmost object. Object 4 is the topmost object. The robot is not holding anything. In the goal, Object 2 is above object 3. Object 4 is above object 2. Object 3 is on Object 1. Object 4 is the topmost object. The robot is not holding anything. Choose a sequence of actions to accomplish this task, and return the action sequence following the example output provided.

LLM: pick(4, 1, 0), stack(4, 2, 0), pick(2, 1, 0), stack(2, 3, 0).

Prompt: Your plan failed in execution, please generate a different one. This may involve sample new plans or reorder the last plan. Please generate output step-by-step, which includes your reasoning for the failure of the last plan. Answer the questions: (i) what is the cause of the failure of the last plan? (ii) do you see similar mistakes in other steps in the plan? Here are the failure reasons: Motion planning failed in pick(2, 1, 0) because of collision with Object 4.

LLM: Got it! Here are my analysis to the planning failures…..

Prompt: Now, based on your above failure reasoning for (i) and (ii) , generate a new task plan to solve the original planning problem. Only return the sequence of logical actions following the format of example output.

LLM: pick(2, 1, 0), stack(2, 3, 0), pick(4, 1, 0), stack(4, 2, 0).

---

### B.3.6 RULEASMEM

The overall template of RuleAsMem is very similar to that of LLMTAMP. However, in RuleAsMem, the planning domain is provided and the problem is defined in PDDL syntax.

**Part 1. Initialization:**

*Imagine you are a robot arm operator, you need to generate a sequence of actions to achieve the given goal. Here is the PDDL planning domain: PDDL planning domain*

**Part 2. Human demonstrations:**

*Here are some examples for you to learn:*

*example of input and output of the system*

**Part 3. New planning problem:**

*Now you are given a new input planning problem as the following: The initial state and the goal of the problem in PDDL syntax. Choose a sequence of actions to accomplish this task. Only return the sequence of logical actions following the format of example output.*

---

**Example for a stacking problem**

Prompt: Imagine you are a robot arm operator, you need to generate a sequence of actions to achieve the given goal. Here is the PDDL planning domain:

```
(define (domain LLM_generated_domain)
    (:requirements :strips :equality)
    (:predicates
        (box ?b1)
        (table ?t1)
        (above ?b1 ?b2)
        (holding ?b1 ?r1)
        (robot ?r1)
                                        ......
```

Here are some examples for you to learn:
Example input:
Initial state: (top, 2), (box, 2), (on_table, 2, 1), (top, 3), (box, 3), (on_table, 3, 1), (top, 4), (box, 4), (on_table, 4, 1), (table, 1), (not_holding, 0), (robot, 0)
Goal state: (top, 2), (box, 2), (above, 2, 3), (box, 3), (above, 3, 4), (box, 4), (on_table, 4, 1), (table, 1), (not_holding, 0), (robot, 0)
Example Output: pick(3, 1, 0), stack(3, 4, 0), pick(2, 1, 0), stack(2, 3, 0).

Now you are given a new planning problem as the following:
Initial state: (top, 2), (box, 2), (on_table, 2, 1), (top, 3), (box, 3), (on_table, 3, 1), (top, 4), (box, 4), (on_table, 4, 1), (table, 1), (not_holding, 0), (robot, 0)
Goal state: (top, 4), (box, 4), (above, 4, 2), (box, 2), (above, 2, 3), (box, 3), (on_table, 3, 1), (table, 1), (not_holding, 0), (robot, 0)
Choose a sequence of actions to accomplish this task, and return the action sequence following the example output provided.

LLM: pick(2, 1, 0), stack(2, 3, 0), pick(4, 1, 0), stack(4, 2, 0).

---

### B.4 EXPERIMENT TASKS

### B.4.1 STACKING

The stacking task involves collecting individual objects and placing them on top of each other to form stable stacks.

### B.4.2 UNSTACKING

The unstacking task is the inverse process of stacking, requiring the robot to identify, grasp, and remove items from existing stacks without disturbing surrounding structures.

### B.4.3 REARRANGEMENT

The rearrangement task demands the robot to relocate objects from an initial configuration into a desired layout.

### B.4.4 ALIGNMENT

The alignment task requires the robot to position multiple objects in a straight line with consistent spacing and orientation.

### B.4.5 COLOR CLASSIFICATION

In the color classification task, the robot must identify the color of each object, group them by color category, and stack or place them in designated areas accordingly.

### B.4.6 PARTS ASSEMBLY

The parts assembly task involves recognizing components and sequentially assembling multiple machining parts together.

### B.4.7 TOWER OF HANOI

The Tower of Hanoi is a puzzle that involves moving a stack of disks from one base to another, one at a time, without ever placing a larger disk on top of a smaller one, using a third base as an auxiliary.

### B.4.8 BRIDGE BUILDING

In bridge building, the robot is required to collect distributed blocks and configure it into a bridge structure.

### B.4.9 BURGER COOKING

Lastly, for burger cooking, the robot needs to stack and pack the food ingredients together to make hamburgers.

## B.5 GENERALIZATION ACROSS TASK COMPLEXITY

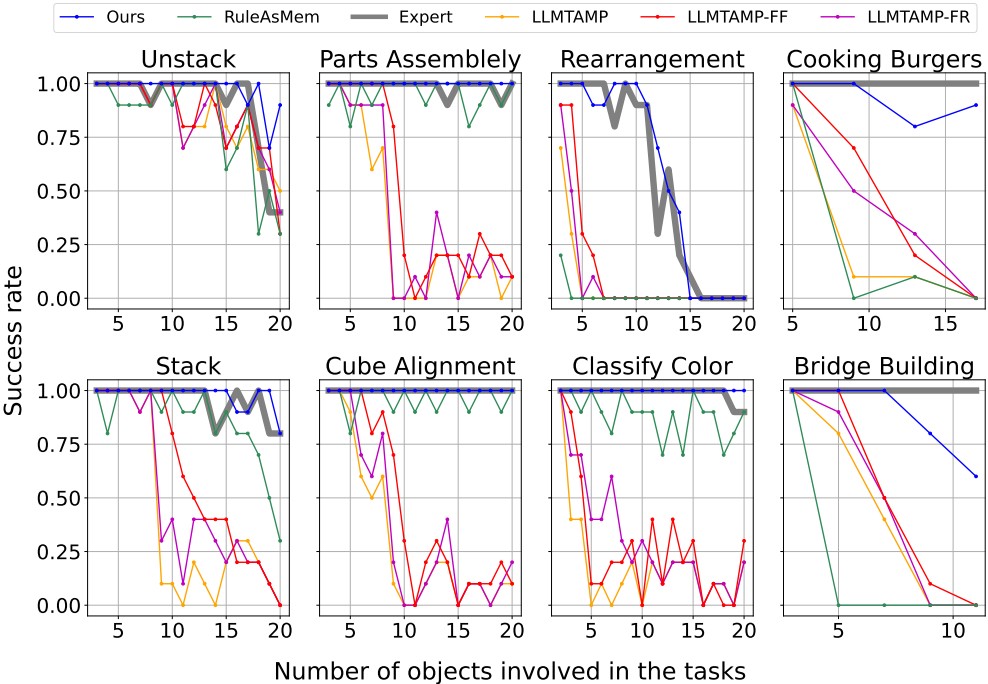

Figure 10: Planning success rate trend across increasing object counts for each task.

### B.6 TIME COST OF LLMTAMP REASONING MODEL VARIANTS

As noted in the paper, the reasoning models incur substantial computational overhead when generating task plans. To account for this, we extended the time limits for **o1-TAMP** and **R1-TAMP** in order to evaluate the contribution of reasoning to planning performance and to enable a fair comparison with our method. This section presents a comprehensive comparison of the time costs between the LLMTAMP variants and our approach.

As shown in Table 6, our method consistently yields lower average planning times across all tasks compared to the LLMTAMP reasoning model variants. This improvement is primarily attributed to PDDLLM's ability to structurally summarize the reasoning process into a standardized planning domain during a one-time offline inference step. As a result, no additional reasoning is required at test time. In contrast, LLMTAMP variants conduct reasoning independently for each task instance, leading to significantly higher computational costs. Such overhead makes these models impractical for deployment on physical robots, where real-time planning capabilities are often essential. As shown in Table 2, while **o1-TAMP** achieves comparable performance in certain tasks, the substantially higher time cost undermines its practical value, particularly in real robotic scenarios.

Table 6: Comparison of time cost between PDDLLM and LLMTAMP reasoning variants

| Experiment | Ours | o1-TAMP | o1-TAMP |
|---|---|---|---|
| **Average Planning Time Cost (Second)** | | | |
| Stack | 5.94 | 39.22 | 211.40 |
| Rearrangement | 15.74 | 92.83 | 337.10 |
| Tower of Hanoi | 4.29 | 167.82 | 353.88 |
| Bridge Building | 6.45 | 82.47 | 305.07 |

### B.7 PROMPT VARIATION TEST

In addition to the five main research questions discussed in the paper, we further evaluated the robustness of our method under variations in the prompting styles. Four different cases were tested to assess the stability of domain generation: parallel prompting with varying numbers of prompts, altering the prompting sequence of simulation outcomes, and tuning the prompting template.

#### B.7.1 PARALLEL PROMPTING FOR PDDLLM

We perform parallel prompting of the LLM multiple times to obtain multiple responses simultaneously. These outputs are subsequently analyzed and aggregated to synthesize an optimal solution. The core prompting procedure follows the same structure as described in Section B.3.1. **All the domains failed to execute are removed from the candidate pool**. Among those successful planning domain candidates, additional post-processing steps are applied to select, summarize and consolidate the results. For predicate naming, we further prompt the LLM to select the most suitable name from among the parallel outputs. For predicate selection, the prompt template is shown as the following:

*Here is a list of predicate describing the same object state: predicate list. Please choose the PDDL predicate from the provided ones to best describe the scenario. Return the chosen one in PDDL syntax.*

Similarly, for optimal actions selection:

*Here is a list of actions describing the same robot skill: action list. Please choose the PDDL action from the provided ones to best describe the scenario. Return the chosen one in PDDL syntax.*

For relevance classification, if the parallel outputs are inconsistent, PDDLLM selects the majority response. In the case of a tie, it randomly selects one among the tied options.

---

**Example for a choosing predicate from 5 parallel outputs**

Prompt: Here is a list of predicate describing the same object state: [(above, ?o1, ?o2), (is_on, ?o1, ?o2), (aligned_vertically, ?o1, ?o2), (on_top_of, ?o1, ?o2), (above_object, ?o1, ?o2) ]. Please choose the PDDL predicate from the provided ones to best describe the scenario. Return the chosen one in PDDL syntax.

LLM: (above, ?o1, ?o2)

---

**Example for a choosing predicate from 10 parallel outputs**

Prompt: Here is a list of predicate describing the same object state: [(above, ?o1, ?o2), (is_on, ?o1, ?o2), (aligned_vertically, ?o1, ?o2), (on_top_of, ?o1, ?o2), (above_object, ?o1, ?o2), (above, ?o1, ?o2), (on, ?o1, ?o2), (vertically_on, ?o1, ?o2), (on_top, ?o1, ?o2), (upper, ?o1, ?o2)]. Please choose the PDDL predicate from the provided ones to best describe the scenario. Return the chosen one in PDDL syntax.

LLM: (on, ?o1, ?o2)

---

#### B.7.2 ALTERING THE PROMPTING SEQUENCE OF SIMULATION OUTCOMES

In this prompt variation, we alter the ordering of dimensions presented to the LLM. While the prompt template remains similar to that described in Section B.3.1, we do not adhere to a fixed dimension sequence. Instead, we shuffle the order of dimensions within the prompt to evaluate whether this affects the generated output. Some examples are provided here.

---

**Example of prompting object position relation predicate with altered sequence - A**

Prompt: There are two objects in the environment, whose positions are $(x_1, y_1)$ and $(x_2, y_2)$. In the horizontal direction, $|x_1 - x_2| \leq u$. In the vertical direction, $u < y_1 - y_2 \leq 2u$. Please create a predicate in PDDL syntax to describe this relation and classify if it is related to the task of stacking cubes together. Please return the result in the following format: predicate, relevance.
LLM: is_on($? o_1$, $? o_2$), related.

---

**Example of prompting object position relation predicate with altered sequence - B**

Prompt: There are two objects in the environment, whose positions are $(x_1, y_1)$ and $(x_2, y_2)$. In the vertical direction, $u < y_1 - y_2 \leq 2u$. In the horizontal direction, $|x_1 - x_2| \leq u$. Please create a predicate in PDDL syntax to describe this relation and classify if it is related to the task of stacking cubes together. Please return the result in the following format: predicate, relevance.
LLM: is_on($? o_1$, $? o_2$), related.

---

### B.7.3 TEMPLATE TUNING

In this experiment, we aim to evaluate whether slight modifications to the prompt template affect the LLM's ability to generate predicate names and assess their relevance. Here is the fine-tuned template:

*There are n objects in the environment, whose feature name are feature value 1 and feature value 2. In dimension 1, we know feature subspace range in dimension 1. In dimension 2, we know feature subspace range in dimension 2. Please create a predicate in PDDL syntax to describe this relation. Assign a score to this predicate indicating its relevance to the task of current task description. The score range is 0 to 1, where 0 indicate irrelevant and 1 indicate very relevant. Please return the result in the following format: predicate, score.*

After collecting the scores, we set a threshold to determine which predicates are relevant. In the experiment, the threshold chosen is 0.5.

---

**Example of prompting object position relation predicate with modified template**

Prompt: There are two objects in the environment, whose positions are $(x_1, y_1)$ and $(x_2, y_2)$. In the horizontal direction, we know $|x_1 - x_2| \leq u$. In the vertical direction, we know $u < y_1 - y_2 \leq 2u$. Please create a predicate in PDDL syntax to describe this relation. Assign a score to this predicate indicating its relevance to the task of stacking cubes together. The score range is 0 to 1, where 0 indicate irrelevant and 1 indicate very relevant. Please return the result in the following format: predicate, score.
LLM: above($? o_1$, $? o_2$), 0.9.

---

The results, presented in Table 7, demonstrate that our method remains robust across different prompting styles. Regardless of the prompt variation, the generated planning domains consistently solve the test tasks with a success rate approaching 100%. Minor fluctuations are attributed to randomness in the planning search process and occasional motion execution failures.

Table 7: Planning success rate for domains generated using different prompt styles.

| Experiment | 10-Parallel | 5-Parallel | Sequence Altering | Template Tuning |
|---|---|---|---|---|
| Stack | 97.8% | 96.1% | 96.1% | 95.6% |
| Unstack | 97.8% | 100% | 98.9% | 98.3% |
| Cube Alignment | 100% | 100% | 100% | 100% |

## B.8 PLANNING TIME LIMIT VARIATION

Table 8: Planning success rate (%) across tasks for all methods (Time limit = 25 s).

| Method | Expert | LLMTAMP | LLMTAMP-FF | LLMTAMP-FR | RuleAsMem | PDDLLM |
|---|---|---|---|---|---|---|
| Stack | $95.8 \pm 0.2$ | $41.5 \pm 4.3$ | $56.8 \pm 3.4$ | $43.8 \pm 3.8$ | $84.3 \pm 3.3$ | $97.5 \pm 1.6$ |
| Unstack | $99.4 \pm 0.6$ | $81.2 \pm 1.1$ | $85.0 \pm 5.7$ | $85.5 \pm 5.4$ | $79.8 \pm 1.9$ | $94.9 \pm 0.5$ |
| Color Classification | $96.3 \pm 0.1$ | $18.1 \pm 1.5$ | $24.9 \pm 0.8$ | $23.1 \pm 3.3$ | $87.6 \pm 1.9$ | $99.5 \pm 0.4$ |
| Alignment | $100.0 \pm 0.0$ | $31.6 \pm 3.1$ | $40.9 \pm 2.0$ | $35.3 \pm 2.0$ | $96.0 \pm 0.8$ | $100.0 \pm 0.0$ |
| Parts Assembly | $98.9 \pm 0.6$ | $33.6 \pm 1.5$ | $46.1 \pm 2.1$ | $37.5 \pm 1.4$ | $95.1 \pm 0.6$ | $100.0 \pm 0.0$ |
| Rearrange | $67. \pm 0.6$ | $5.6 \pm 1.1$ | $11.7 \pm 0.6$ | $7.4 \pm 1.1$ | $1.1 \pm 0.6$ | $52.5 \pm 2.2$ |
| Burger Cooking | $100.0 \pm 0.0$ | $27.8 \pm 2.8$ | $45.1 \pm 7.3$ | $38.9 \pm 5.6$ | $27.8 \pm 2.8$ | $89.6 \pm 3.2$ |
| Bridge Building | $100.0 \pm 0.0$ | $43.3 \pm 3.3$ | $48.9 \pm 5.9$ | $47.2 \pm 4.3$ | $20.0 \pm 0.0$ | $87.2 \pm 4.3$ |
| Tower of Hanoi | $83.3 \pm 2.4$ | $14.3 \pm 0.0$ | $14.3 \pm 0.0$ | $14.3 \pm 0.0$ | $14.3 \pm 0.0$ | $85.7 \pm 0.0$ |
| **Overall** | $93.2 \pm 0.2$ | $34.5 \pm 0.4$ | $43.2 \pm 0.8$ | $38.1 \pm 0.9$ | $68.3 \pm 0.9$ | $90.5 \pm 0.9$ |

Table 9: Planning success rate (%) across tasks for all methods (Time limit = 100 s).

| Method | Expert | LLMTAMP | LLMTAMP-FF | LLMTAMP-FR | RuleAsMem | PDDLLM |
|---|---|---|---|---|---|---|
| Stack | $99.5 \pm 0.5$ | $41.5 \pm 4.3$ | $76.7 \pm 2.7$ | $71.0 \pm 2.7$ | $85.5 \pm 2.9$ | $97.5 \pm 1.6$ |
| Unstack | $96.1 \pm 0.2$ | $90.3 \pm 1.5$ | $96.9 \pm 1.2$ | $96.1 \pm 1.1$ | $88.4 \pm 1.2$ | $97.7 \pm 0.7$ |
| Color Classification | $100.0 \pm 0.0$ | $18.1 \pm 1.5$ | $42.0 \pm 1.9$ | $64.0 \pm 2.2$ | $88.7 \pm 2.3$ | $100.0 \pm 0.0$ |
| Alignment | $100.0 \pm 0.0$ | $31.6 \pm 3.1$ | $55.7 \pm 3.8$ | $44.3 \pm 3.8$ | $96.0 \pm 0.8$ | $100.0 \pm 0.0$ |
| Parts Assembly | $98.9 \pm 0.6$ | $33.6 \pm 1.5$ | $57.1 \pm 0.8$ | $47.7 \pm 2.5$ | $95.1 \pm 0.6$ | $100.0 \pm 0.0$ |
| Rearrange | $73.9 \pm 0.2$ | $5.6 \pm 1.1$ | $17.4 \pm 1.1$ | $14.7 \pm 1.2$ | $1.1 \pm 0.6$ | $69.4 \pm 0.5$ |
| Burger Cooking | $100.0 \pm 0.0$ | $27.8 \pm 2.8$ | $50.0 \pm 4.8$ | $51.4 \pm 6.1$ | $27.8 \pm 2.8$ | $97.2 \pm 2.8$ |
| Bridge Building | $100.0 \pm 0.0$ | $43.3 \pm 3.3$ | $53.3 \pm 3.8$ | $57.8 \pm 2.2$ | $20.0 \pm 0.0$ | $87.2 \pm 4.3$ |
| Tower of Hanoi | $100.0 \pm 0.0$ | $14.3 \pm 0.0$ | $14.3 \pm 0.0$ | $14.3 \pm 0.0$ | $14.3 \pm 0.0$ | $100.0 \pm 0.0$ |
| **Overall** | $95.9 \pm 0.1$ | $35.8 \pm 0.4$ | $55.6 \pm 1.2$ | $54.8 \pm 0.7$ | $69.9 \pm 0.7$ | $94.2 \pm 0.5$ |

As outlined in the experimental design section, we set a default planning time limit of 50 seconds to reflect the real-time constraints commonly imposed on robotic systems. Although prior studies typically allow planning times on the order of minutes (Silver et al., 2021; Khodeir et al., 2023; Huang et al., 2025), the exact limits vary considerably. To assess the robustness of our approach under different time allowances, we evaluate performance at time limits of 25, 50, and 100 seconds. The results show that our method consistently outperforms all LLM-based baselines across all tested time settings and task types. Moreover, the planning domains derived by PDDLLM yield performance comparable to that of expert-designed domains under all time settings and across all tasks. All simulations were conducted on a system with an Intel Core i9-14900KF CPU without GPU acceleration, and all LLM prompting was performed using API services.

## B.9 EXAMPLE OF PDDLLM IMAGINED PREDICATES

Table 10: Examples of imagined predicates across multiple categories.

**Predicate Name**

**Object Position Predicates**

```
(above ?a ?b)                              (forall ?a (above ?a ?b))
(forall ?a (above ?a ?b) is false)         (not (above ?a ?b))
(beside ?a ?b)                             (forall ?a (beside ?a ?b))
(forall ?a (beside ?a ?b) is false)        (not (beside ?a ?b))
(front ?a ?b)                              (forall ?a (front ?a ?b))
(forall ?a (front ?a ?b) is false)         (not (front ?a ?b))
(far-x ?a ?b)                              (forall ?a (far-x ?a ?b))
(forall ?a (far-x ?a ?b) is false)         (not (far-x ?a ?b))
(apart-y ?a ?b)                            (forall ?a (apart-y ?a ?b))
(forall ?a (apart-y ?a ?b) is false)       (not (apart-y ?a ?b))
(distant-x ?a ?b)                          (forall ?a (distant-x ?a ?b))
(forall ?a (distant-x ?a ?b) is false)     (not (distant-x ?a ?b))
```

**Object Support Predicates**

```
(on-table ?a ?t)                           (not (on-table ?a ?t))
(forall ?a (on-table ?a ?t))               (forall ?a (on-table ?a ?t) is false)
(aligned-x ?a ?t)                          (not (aligned-x ?a ?t))
(forall ?a (aligned-x ?a ?t))              (forall ?a (aligned-x ?a ?t) is false)
(aligned-y ?a ?t)                          (not (aligned-y ?a ?t))
(forall ?a (aligned-y ?a ?t))              (forall ?a (aligned-y ?a ?t) is false)
(above-table ?a ?t)                        (not (above-table ?a ?t))
(forall ?a (above-table ?a ?t))            (forall ?a (above-table ?a ?t) is false)
(near-x ?a ?t)                             (not (near-x ?a ?t))
(forall ?a (near-x ?a ?t))                 (forall ?a (near-x ?a ?t) is false)
(far-x ?a ?t)                              (not (far-x ?a ?t))
(forall ?a (far-x ?a ?t))                  (forall ?a (far-x ?a ?t) is false)
(far-y ?a ?t)                              (not (far-y ?a ?t))
(forall ?a (far-y ?a ?t))                  (forall ?a (far-y ?a ?t) is false)
(near-each ?a ?t)                          (not (near-each ?a ?t))
(forall ?a (near-each ?a ?t))              (forall ?a (near-each ?a ?t) is false)
```

**Robot Predicates**

```
(holding ?a ?r)                            (not (holding ?a ?r))
(forall ?a (holding ?a ?r))                (forall ?a (holding ?a ?r) is false)
(gripper-near-open ?a ?r)                  (not (gripper-near-open ?a ?r))
(forall ?a (gripper-near-open ?a ?r))      (forall ?a (gripper-near-open ?a ?r) is false)
(gripper-far ?a ?r)                        (not (gripper-far ?a ?r))
(forall ?a (gripper-far ?a ?r))            (forall ?a (gripper-far ?a ?r) is false)
(gripper-near ?a ?r)                       (not (gripper-near ?a ?r))
(forall ?a (gripper-near ?a ?r))           (forall ?a (gripper-near ?a ?r) is false)
```

**Color Predicates**

```
similar-color ?a ?b                            (not similar-color ?a ?b)
(forall ?a similar-color ?a ?b)                (forall ?a similar-color ?a ?b is false)
moderate-color-difference ?a ?b                (not moderate-color-difference ?a ?b)
(forall ?a moderate-color-difference ?a ?b)    (forall ?a moderate-color-difference ?a ?b is false)
distinct-colors ?a ?b                          (not distinct-colors ?a ?b)
(forall ?a distinct-colors ?a ?b)              (forall ?a distinct-colors ?a ?b is false)
very-different-color ?a ?b                     (not very-different-color ?a ?b)
(forall ?a very-different-color ?a ?b)         (forall ?a very-different-color ?a ?b is false)
```

**Size Predicates**

```
(smaller ?a ?b)                            (not (smaller ?a ?b))
(forall ?a (smaller ?a ?b))                (forall ?a (smaller ?a ?b) is false)
(larger ?a ?b)                             (not (larger ?a ?b))
(forall ?a (larger ?a ?b))                 (forall ?a (larger ?a ?b) is false)
(longer ?a ?b)                             (not (longer ?a ?b))
(forall ?a (longer ?a ?b))                 (forall ?a (longer ?a ?b) is false)
(taller ?a ?b)                             (not (taller ?a ?b))
(forall ?a (taller ?a ?b))                 (forall ?a (taller ?a ?b) is false)
(wider ?a ?b)                              (not (wider ?a ?b))
(forall ?a (wider ?a ?b))                  (forall ?a (wider ?a ?b) is false)
(taller-wider ?a ?b)                       (not (taller-wider ?a ?b))
(forall ?a (taller-wider ?a ?b))           (forall ?a (taller-wider ?a ?b) is false)
(wider-longer ?a ?b)                       (not (wider-longer ?a ?b))
(forall ?a (wider-longer ?a ?b))           (forall ?a (wider-longer ?a ?b) is false)
```

Table 11: Examples of predicate generation across different task and scenes.

| | |
|---|---|
| **Predicates for Weight** | |
| Lift a box in warehouse | `(light-box ?a): 40 < weight < 70 kg`
`(heavy-box ?a): 70 < weight < 100 kg` |
| Collect apples of different sizes | `(small-apple ?a): 100 < w < 166.7 g`
`(medium-apple ?a): 166.7 < w < 233.3 g`
`(large-apple ?a): 233.3 < w < 300 g` |
| **Predicates for Length** | |
| Fulfill a 15-cm coffee cup | `(needs-refill ?a): 0 < level < 7.5 cm`
`(nearly-full ?a): 7.5 < level < 11.25 cm` |
| Measure high school students' heights | `(medium-height ?a): 150 < h < 160 cm`
`(average-height ?a): 160 < h < 170 cm`
`(tall-height ?a): 170 < h < 180 cm`
`(very-tall ?a): 180 < h < 190 cm` |
| **Predicates for Air Quality** | |
| Plan when to travel | `(suitable-travel ?a): 0 < PM2.5 < 166.7`
`(high-risk ?a): 333.3 < PM2.5 < 500` |
| **Predicates for Brightness** | |
| Fix broken light in office | `(dimly-lit ?a): 0 < brightness < 150 lux` |
| **Predicates for Temperature** | |
| Heat a kitchen stove | `(stove-hot ?a): 110 < T < 200 °C` |
| Boil milk | `(cold-milk ?a): 0 < T < 10 °C`
`(warm-milk ?a): 20 < T < 40 °C`
`(boiling-milk ?a): 90 < T < 100 °C` |

B.10   Test Problem Set Design

B.10.1   Dimensions of Difficulty

**Task Diversity:** To ensure robustness and broad applicability, we sampled over 1,200 planning tasks across nine distinct environments, including stacking, unstacking, rearrangement, alignment, parts assembly, color classification, burger cooking, and bridge building. The details of the tasks are provided in Section B.4. Each task category includes tasks of 3 to 20 objects, with 10 distinct tasks sampled for each object count. The resulting task plan lengths ranged from 6 to 510 steps, reflecting a wide spectrum of planning horizons. Our experiments spanned over 150 unique predicates.

**Task Complexity:** In task design, we consider two types of complexity: domain derivation complexity and planning complexity. *Domain derivation complexity* is determined by the number of predicates imagined and actions invented by PDDLLM; the more predicates and actions, the higher the complexity. *Planning complexity* is influenced by both the planning domain and the task. Given $n$ objects, a task plan of length $l$, and $m$ actions in the domain, the branching factor at each step is $m \times n$, resulting in a approximate complexity of $(m \times n)^l$. In Section B.10, we layout a table showcase the planning complexity (in approximate order of magnitude) of the most difficult problem in the category and the domain derivation complexity. The Tower of Hanoi task exhibits the greatest planning complexity, while bridge building presents the highest domain derivation complexity. In contrast, stack and unstack are simpler in both complexity measures.

**Knowledge Transferability:** We evaluate the knowledge transferability of PDDLLM by testing its ability to generalize from demonstrated tasks to novel ones with overlapping predicates and actions. For most tasks, PDDLLM was given demonstrations of the same task involving fewer (3 to 4) objects. However, for compositional tasks such as rearrangement and bridge building, the model was instead provided demonstrations of simpler subtasks. Specifically, rearrangement used demonstrations from both stacking and unstacking, while bridge building combined demonstrations of stacking and alignment. These setups test whether PDDLLM can compose skills learned from simpler demonstrations to solve more complex tasks.

Table 12: Maximum planning complexity and domain derivation complexity of each category .

| Task | Stack | Unstack | Color Classify | Alignment | Parts Assembly | Rearrange | Burger Cook | Bridge Build | Tower of Hanoi |
|---|---|---|---|---|---|---|---|---|---|
| **Max Planning Complexity** | $10^{58}$ | $10^{58}$ | $10^{71}$ | $10^{58}$ | $10^{58}$ | $10^{137}$ | $10^{48}$ | $10^{36}$ | $10^{307}$ |
| **Domain Derivation Complexity** | 90 | 90 | 111 | 94 | 92 | 96 | 114 | 128 | 94 |

B.10.2   Derivation of Bounded Domain Derivation Complexity

The predicate imagination complexity measures how many predicates need to be generated. Our method is generalizable to more complex cases for the following reasons:

**Complexity calculation formula**   The formula is:

$$\text{complexity} = \sum_i n_{p,i}^{n_{\dim,i}}$$

where $n_{\dim,i}$ is the number of dimensions included in predicate type $i$. For example, $n_{\dim,\text{color}} = 3$ since color is represented by the three RGB channels. $n_{p,i}$ is the number of partitions along each dimension for predicate type $i$. The term $n_{p,i}^{n_{\dim,i}}$ gives the total number of predicates generated for type $i$, and summing over all predicate types yields the overall complexity of the predicate imagination process for the task.

**The Method is Scalable as the Increase in Predicate Types Leads to Linear Growth**   The number of predicate types grow linearly as the task complexity increases. The only exponential factor

comes from the number of dimensions $n_{\text{dim}}$ per predicate type; however, in practice, this value is typically small (e.g., $\leq 3$).

**High-Dimensional Features Can Be Decomposed to Reduce Complexity**    For higher-dimensional features, as object pose $(x, y, z, \text{yaw}, \text{pitch}, \text{roll})$, we can decompose them into lower-dimensional predicate types (e.g., position and orientation). This reduces exponential complexity to additive linear terms, preserving scalability and enabling rich predicate representation.

**Task Planning Does Not Require Many Partitions**    Dividing features into too many partitions makes predicate implication nearly continuous, which defeats the purpose of symbolic abstraction. Here is an experiment:

- **2 partitions:** "cold" ($0 < T < 50°C$), "hot" ($50 < T < 100°C$)
- **4 partitions:** "cool" ($0 < T < 25°C$), "warm" ($25 < T < 50°C$), "hot" ($50 < T < 75°C$), "near-boiling" ($75 < T < 100°C$)
- **100 partitions:** ..., "post-boiling" ($95 < T < 96°C$), "stable-boil" ($96 < T < 97°C$), ...

For the same task "boil milk", we compare the predicate imagination results by forcing the system to partition the feature space into 2, 4, and 100 intervals. It is evident that overly fine partitioning reduces interpretability and semantic efficiency, so the number of partitions $n_p$ per dimension is usually low.

### B.11    MISSING AND REDUNDANT ELEMENTS

A redundant predicate or action refers to one that, when removed from the planning domain, does not degrade planning performance. Conversely, a missing predicate or action is one whose absence significantly reduces the domain's planning performance.

The reported percentages are computed by comparing the PDDLLM generated PDDL domains with the expert-designed PDDL domains. The percentage of missing predicates/actions reflects the completeness of the generated domain (i.e., whether all necessary components are present). The percentage of redundant predicates/actions reflects the optimality of the domain (i.e., whether unnecessary components have been excluded).

### B.12    REAL-WORLD DEMONSTRATION COLLECTION

To enable the demonstration collection in real-world, we build up a data collection system using Agilex Pika and ArUco markers. We collect both the end-effector trajectories of the human demonstrator and the pose of the target objects. These data are later mapped to symbolic predicates and actions in the PDDL domain, providing a bridge between raw demonstration data and high-level planning representations.

**End-Effector Trajectory:** The movement of the human operator's end effector is recorded using the Agilex Pika Data Collection System. The system's motion capture capability provides precise position and orientation measurements over time. These trajectories serve as a basis for identifying meaningful segments of motion that can be abstracted into candidate actions for the symbolic domain.

**Object Properties and States:** Physical characteristics of the objects, such as size, are obtained from the available CAD models, ensuring that the symbolic model is grounded in realistic object dynamics. In addition, the poses of the objects during demonstrations are captured using ArUco markers affixed to their surfaces. These pose estimates allow us to track state changes—such as whether an object has been grasped, moved, or placed—which are subsequently encoded as predicates in the PDDL formulation.

### B.13    PERCEPTION MODULES AND ABLATION STUDY ON NOISE RESISTANCE

**Perception in our real-robot system.** For our real-world experiments, we adopt a ArUco markers as our perception method, which provides reliable object identity and 6D pose estimates. This

setup allows us to evaluate the planning framework without conflating perception errors with the contributions of our method.

**Compatibility with other perception modules.** PDDL task planning is modular and can integrate with other perception systems that provide object state estimation, including learning based (Kase et al., 2020), VLM-based (Liang et al., 2024), or standard 6D object-pose estimation methods (Huang et al., 2025). As our method strictly follows the PDDL pipeline, our approach similarly does not place restrictive assumptions on how these states are produced, as long as a good state estimation is provided.

**Robustness to perception noise.** The logical abstraction produced by our method is formulated as an intersection of intervals, and thus inherently provides tolerance to perception errors: predicates operate on ranges of continuous values rather than exact measurements, which reduces sensitivity to noise. To address your concern, we conduct an additional real world experiment evaluating performance under perturbations of the perceived object state. We randomly place two $4 \text{ cm} \times 4 \text{ cm} \times 2 \text{ cm}$ cubes on a table, use our ArUco marker system to obtain 6D pose estimates, and evaluate each predicate by comparing its predicted truth value against ground-truth spatial relationships inferred from the poses. For each predicate, we construct 50 balanced trials, 25 cases where the predicate should be true and 25 where it should be false. The results are as follows:

| **Predicates** | `is_on` | `next_to` |
|---|---|---|
| Accuracy | 94% | 96% |

In addition, we provide another experiment where we intentionally inject perception noise varying from 5% to 30% to imitate perception methods with varied perception accuracy. Each trial is repeated three times, leading to the following results:

Table 13: Accuracy under injected perception noise (percentage noise levels).

| **Percentage Noise** | **5%** | **10%** | **15%** | **20%** | **25%** | **30%** |
|---|---|---|---|---|---|---|
| `is_on` Accuracy | $92.7 \pm 0.9$ | $88.7 \pm 2.5$ | $84.0 \pm 3.3$ | $80.7 \pm 3.4$ | $80.7 \pm 1.9$ | $78.7 \pm 0.9$ |
| `next_to` Accuracy | $94.7 \pm 0.9$ | $84.0 \pm 1.6$ | $80.0 \pm 4.9$ | $76.0 \pm 4.3$ | $74.0 \pm 2.8$ | $69.3 \pm 1.9$ |
| **Overall Accuracy** | $93.7 \pm 1.4$ | $86.3 \pm 3.1$ | $82.0 \pm 4.6$ | $78.3 \pm 4.5$ | $77.3 \pm 4.1$ | $74.0 \pm 4.9$ |

The experiment shows that the learned predicates exhibit strong tolerance to perception noise, with significant degradation occurring only when injected noise exceeds 20%. This demonstrates the feasibility of integrating our system with current perception technologies.

## B.14    APPLICABLE SCENARIOS ANALYSIS

**Suitable Domains** The proposed system is naturally suited to domains that align with the classical task and motion planning (TAMP) setting (Garrett et al., 2021; Kaelbling & Lozano-Pérez, 2011), namely those with: (1) consistent object physical attributes, (2) scene geometry can be represented with structured models such as URDF/CAD, so that motion-planning constraints can be grounded reliably. Such tasks have already included a wide range of regular robotic tasks, suas table tidying, object sorting, cabinet and drawer organization, simple assembly, etc.(Garrett et al., 2021; Silver et al., 2021; Huang et al., 2025; Silver et al., 2024) These prior studies have largely verified the task and motion planning framework in those domains.

**Challenging Domains** While modern simulators are powerful, we recognize that some domains remain difficult to model reliably, which remains a challenge shared across the robotic planning community. Here we discuss those domains that are specifically challenging to our approach. (1) Deformable, fluid, or highly non-rigid materials (e.g., cloth, liquids), where feature definition is under studied and reliable physics simulation is still limited. (2) Tasks requiring fine-grained feature modeling, such as tight-tolerance insertion, dexterous geometry manipulation, where small errors in simulation yield large discrepancies in behavior. (3) Unstructured or rapidly changing environments, where object properties or contacts vary unpredictably and cannot be captured with static models.

**How our method remain helpful in challenging situations with complex dynamics?**

There are two main aspects:

- **Easing human workload by automating common-dynamics predicate generation.** In challenging domains, the planning domain produced by our approach still provides a strong starting point. Although it may not fully capture the most complex dynamics, it reliably constructs predicates that describe common and regularly occurring dynamics. This leaves only a small number of highly complex predicates for engineers to refine, avoiding the need to rebuild the entire domain from scratch and substantially reducing manual effort.

- **Empirical compatibility with manipulation techniques designed for complex dynamics.** Our method can be integrated with stronger low-level manipulation modules such as VLA, which are specifically engineered to handle complex dynamics. We also include a table-setting experiment where our framework is combined with a finetuned version of Pi0 as the low-level skill controller, achieving a 7/10 success rate in real-robot trials.

### B.15 ABLATION STUDY: HYPERPARAMETER $u$

For subspace granularity hyperparameter $u$, we perform an ablation study on the selection of $u$. Specifically for u selection, for each continuous feature $f$, $u_f$ has to be smaller than the **smallest non-zero observed difference** $d_{min}$ of that feature among all relevant objects. This provides a conservative upper bound of $u_f$, otherwise the system cannot capture distinctions of all objects. Thus, we ablates the selection of u from $d_{min}$ to $0.25d_{min}$. We fix $n_{prompt} = 10$ for all experiments. Three runs were performed for per $u_f$ per task, and the average missing/redundant percentage compared to human design is reported.

Table 14: Effect of the threshold $u$ on redundant and missing predicates across tasks.

| Tasks | Error Type | $u_f = d_{min}$ | $u_f = 0.75d_{min}$ | $u_f = 0.5d_{min}$ | $u_f = 0.25d_{min}$ |
|---|---|---|---|---|---|
| Stacking | Redundant | 8.3% | 12.5% | 16.7% | 12.5% |
| Stacking | Missing | 4.2% | 0.0% | 0.0% | 4.2% |
| Tower of Hanoi | Redundant | 14.3% | 14.3% | 9.5% | 14.3% |
| Tower of Hanoi | Missing | 0.0% | 0.0% | 0.0% | 4.2% |
| Bridge Building | Redundant | 3.7% | 11.1% | 14.8% | 22.2% |
| Bridge Building | Missing | 22.2% | 22.2% | 22.2% | 22.2% |
| Burger Cooking | Redundant | 3.7% | 7.4% | 18.5% | 22.2% |
| Burger Cooking | Missing | 22.2% | 22.2% | 25.9% | 22.2% |
| Overall | Redundant | 7.5% | 11.3% | 14.9% | 17.8% |
| Overall | Missing | 12.1% | 11.1% | 12.0% | 13.2% |

To conclude, reducing $u$ overall would increase the risk of redundant predicates, but the negative effect on missing predicate is less apparent. Overall, the predicate generation is not very sensitive to the variation of $u$. Usually it is also not necessary to set a extremely small $u$. In our implementation, we allow LLM to adjust $u$. Specifically, the LLM can **merge** intervals that are semantically uninformative for the task into a single broader region, and **subdivide** intervals in feature regions where finer distinctions are necessary.

### B.16 ABLATION STUDY: PARALLEL PROMPT

This Section include ablation study of number of parallel prompt $n_{prompt}$ with respect to (1) the domain generation success rate. (2) domain quality with respect to human design.

#### B.16.1 INFLUENCE ON DOMAIN GENERATION SUCCESS RATE

**Empirically, the system consistently produces an executable planning domain when using five or more parallel prompts.** We evaluate four domains, Stacking, Tower of Hanoi, Bridge Building, and Burger Cooking, which span a wide range of domain-generation complexities. For each value of

$n_{\text{prompt}}$, every task is tested with three independent parallel runs. The overall failure rate of producing an executable planning domain as a function of the number of parallel prompts is reported below.

Table 15: Rate of failure to generate executable domain with varying number of parallel prompt

| **Tasks** | $n_{\text{prompt}} = 1$ | $n_{\text{prompt}} = 5$ | $n_{\text{prompt}} = 10$ | $n_{\text{prompt}} = 15$ | $n_{\text{prompt}} = 20$ |
|---|---|---|---|---|---|
| Overall Failure Rate(%) | 41.7 | 8.3 | 8.3 | 0 | 0 |

### B.16.2 Influence on Domain quality Compared to Human Design

The result of varying $n_{prompt}$ is shown in the table below. We the selection of $u_f = d_{min}$ for all experiments. Three runs were performed for per $n_{prompt}$ per task, and the average missing/redundant percentage compared to human design is reported.

Table 16: Redundant and missing predicate rates across tasks as the number of parallel prompts changes.

| **Tasks** | **Error Type** | $n_{\text{prompt}} = 1$ | $n_{\text{prompt}} = 5$ | $n_{\text{prompt}} = 10$ | $n_{\text{prompt}} = 15$ | $n_{\text{prompt}} = 20$ |
|---|---|---|---|---|---|---|
| Stacking | Redundant | 25.0% | 12.5% | 8.3% | 4.2% | 4.2% |
| Stacking | Missing | 4.2% | 0.0% | 4.2% | 0.0% | 0.0% |
| Tower of Hanoi | Redundant | 38.1% | 9.5% | 14.3% | 14.3% | 9.5% |
| Tower of Hanoi | Missing | 0.0% | 0.0% | 0.0% | 0.0% | 0.0% |
| Bridge Building | Redundant | 33.3% | 11.1% | 3.7% | 3.7% | 0.0% |
| Bridge Building | Missing | 29.6% | 25.9% | 22.2% | 22.2% | 22.2% |
| Burger Cooking | Redundant | 29.6% | 11.1% | 3.7% | 11.1% | 0.0% |
| Burger Cooking | Missing | 29.6% | 22.2% | 22.2% | 22.2% | 22.2% |
| Overall | Redundant | 31.5% | 11.1% | 7.5% | 8.3% | 3.4% |
| Overall | Missing | 15.8% | 12.0% | 12.1% | 11.1% | 11.1% |

### B.17 Effect of Missing Complex Predicates

The complex predicates are missed more frequently when they require deep or repeated operator compositions (e.g., `(all_bases_finished)`), which involves multiple nested `forall` quantifiers and conjunctions). Such structures expand the candidate space combinatorial, making them harder for the LLM to reason the nested logic and identify consistently. In contrast, simpler higher-order predicates, such as `on_top`, which require only shallow or singular compositions, are generated reliably with almost no missing observed. In practice, missing highly composite predicates rarely leads to direct planning failure. The essential geometric and relational structure is already captured by first-order predicates; the more complex, higher-order predicates primarily serve to accelerate planning by enforcing appropriate action ordering. For example, omitting `(all_bases_finished)` in the bridge-building task causes the planner to initiate surface assembly prematurely, encounter motion-planning failure, and then backtrack to assemble the remaining bases. Our experiments still achieve over 85% success even without such predicates—thus, the increased planning time does not cause immediate failure in typical settings. However, this becomes more problematic for time-sensitive tasks. In scenarios with strict latency requirements or limited search budgets, missing these complex predicates may cause the planner to exceed the allotted time, effectively resulting in task failure even though a valid plan is reachable.

### B.18 Refining Predicate Precision with Improved Language Guidance

Here we provide an additional experiment that illustrates how refined language guidance can improve the predicate precision. The only change we made in the following experiment is the task description. All other settings are consistent with the experiments in the main paper.

Table 17: Examples of predicate generation across different task and scenes.

| Task | Predicate Name |
|---|---|
| Heat Milk | `(hot-milk ?a):  40 < T < 60 °C`
`(warm-milk ?a):  20 < T < 40 °C` |
| Boil milk | `(cold-milk ?a):  0 < T < 10 °C`
`(warm-milk ?a):  20 < T < 40 °C`
`(boiling-milk ?a):  90 < T < 100 °C` |
| Heat the cold milk from the fridge until it comes to a boil. | `(cold-milk ?a):  0 < T < 10 °C`
`(near-boil-milk ?a):  80 < T < 90 °C`
`(boiling-milk ?a):  90 < T < 100 °C` |

As shown in the results, vague language prompts lead to vague predicates and loose predicate constraints. When the instruction becomes more specific, the generated predicates also become more precise, with correspondingly sharper constraints.

### B.19    FEEDBACK LOOP FOR LOW-LEVEL MOTION FAILURE

We recognize that low-level execution inaccuracies are indeed a common challenge in task and motion planning methods (Kaelbling & Lozano-Pérez, 2011; Garrett, 2018; Silver et al., 2023). In our framework, we follow the standard task and motion planning paradigm to handle such failures. When a low-level failure occurs (e.g., the third block falls because the second block was not perfectly centered), it result in a logical state change. The PDDL planner then re-invokes symbolic planner to generate a new sequence of actions that recovers from the failure by re-stacking the misplaced block. This replanning–execution loop is repeated until either the goal is achieved or the time limit is reached. This mechanism ensures that occasional geometric inaccuracies do not terminate the task prematurely.

### B.20    TOKEN COST DETAILS

The following table shows the token cost observed in each of 5 trials for three tasks. The average value is reported in the main paper Table 2. Each trail we adopt $\mathbf{u} = \mathbf{d}_{\min}$ and $n_{prompt} = 10$.

Table 18: Token cost (k) across 5 trials for each task in domain derivation.

| Task | Trial 1 | Trial 2 | Trial 3 | Trial 4 | Trial 5 | Avg |
|---|---|---|---|---|---|---|
| Rearrangement | 306.4 | 292.9 | 322.5 | 465.1 | 285.2 | 334.4 |
| Tower of Hannoi | 478.6 | 526.6 | 558.6 | 547.2 | 564.1 | 535.1 |
| Bridge Building | 293.0 | 478.2 | 336.5 | 471.0 | 298.0 | 375.3 |

## B.21 ABLATION STUDY: LLM BACKBONES

To assess whether PDDLLM depends on GPT-4o, we repeated all experiments using Qwen3-4B and Qwen3-8B as backbones. As shown in Table 8, both models achieve performance comparable to GPT-4o across all tasks. The maximum discrepancy across the nine tasks is small (within 2–3% for most tasks), and the average deviation from the GPT-4o results is under 1%. The results are expected, as once a planning domain is derived, the downstream success rate primarily depends on the PDDL solver rather than the choice of LLM backbone.

Table 19: Performance (%) of PDDLLM with different LLM backbones.

| Backbone Model | Stack | Unstack | Color | Align | Parts | Rearr. | Burger | Bridge | ToH | Overall |
|---|---|---|---|---|---|---|---|---|---|---|
| **GPT-4o** | $97.5 \pm 1.6$ | $97.7 \pm 0.7$ | $100 \pm 0.0$ | $100 \pm 0.0$ | $100 \pm 0.0$ | $64.3 \pm 0.7$ | $91.7 \pm 4.8$ | $87.2 \pm 4.3$ | $100 \pm 0.0$ | $93.3 \pm 0.7$ |
| **Qwen3-4B** | $97.7 \pm 1.7$ | $97.5 \pm 1.6$ | $100 \pm 0.0$ | $100 \pm 0.0$ | $100 \pm 0.0$ | $64.0 \pm 1.3$ | $93.1 \pm 3.7$ | $86.1 \pm 3.8$ | $100 \pm 0.0$ | $93.1 \pm 0.4$ |
| **Qwen3-8B** | $97.8 \pm 0.3$ | $100 \pm 0.0$ | $100 \pm 0.0$ | $100 \pm 0.0$ | $98.3 \pm 1.1$ | $65.1 \pm 2.2$ | $95.1 \pm 2.5$ | $85.0 \pm 5.0$ | $100 \pm 0.0$ | $93.6 \pm 0.5$ |

