# OpenReview forum: "One Demo Is All It Takes: Planning Domain Derivation with LLMs from A Single Demonstration"
_ICLR.cc/2026/Conference — ICLR 2026 Poster_

### Official Review · Reviewer_ZvYq · 2025-10-31

**Soundness:** 3
**Presentation:** 3
**Contribution:** 3
**Rating:** 6
**Confidence:** 4

**Summary:**

The paper introduces PDDLLM, a novel framework that enables LLMs to autonomously generate symbolic planning domains from a single human demonstration. PDDLLM derives both automatically by integrating LLM reasoning with physical simulation. Its Logical Constraint Adapter further connects high-level symbolic plans with low-level motion planning, allowing end-to-end execution. Evaluations across over  nine environments demonstrate that PDDLLM outperforms LLM-based baselines in success rate, planning efficiency, and token cost.

**Strengths:**

- PDDLLM automatically derives predicates and actions from a single demonstration without manual engineering or predefined templates.
- Strong empirical results, outperforms six LLM-based baselines across 1,200 tasks.
- Demonstrates effective transfer of learned domains and actions to new, unseen tasks and environments, on multiple robot platforms.

**Weaknesses:**

- The difference from InterPreT is unclear. It seems PDDLLM replaces human-crafted planning examples with a single demonstration and a predefined constraint pool for predicate generation. More clarification on how this differs conceptually or technically would strengthen the paper.
- Is GPT-4o necessary to achieve the reported performance? Have the authors tested other LLMs?
- Since the method lacks a feedback loop, it seems to require perfect constraint design, perfect predicate specification, and perfect action proposals. A single failure at any step could lead to a catastrophic outcome. It is unclear why the method is described as robust and generalizable, could the authors provide examples demonstrating successful performance with imperfect intermediate steps?

**Questions:**

Questions:
- Once a predicate is proposed with constraints, it appears that it is never updated. What happens if the constraints are incorrect? How do you ensure that the hyperparameters within these constraints are generalizable?
- In Figure 2’s illustration, the x, y, and z axes all share the same hyperparameter u. Why is this the case, and how was u selected?
- How do you handle low-level failures? Although the motion planning is formulated as a constrained optimization problem, the inequality constraints can accumulate errors. For example, in the stacking task, the third block may fall if the second block is not perfectly centered on the first one.

Comments:
- L201-203, PDDL clauses have no comma in it

Missing literature:

Zhu et al., PSALM-V: Automating Symbolic Planning in Interactive Visual Environments with Large Language Models. 2025

---

> ### Author Response · Authors · 2025-11-25
> **Reply to W1&2**
>
> Thank you for highlighting the strengths of our automatic domain-construction approach, its empirical performance, and its ability to transfer knowledge to unseen tasks. We appreciate the reviewer’s thoughtful questions, which helped us clarify the method and improve the paper.
>
> We also thank the reviewer for pointing out the missing prior work; we will add this reference in the related-work section *[L90-92]* and discuss how PSALM-V’s feedback-based mechanisms could potentially complement our approach in addressing its limitation *[L510-513]*. We address each of your concerns below.
>
> ## **Weakness 1**
> We would like to clarify the difference on both conceptual and technical levels:
>
> ### **Different sources of knowledge to reduce human reliance:**
> InterPreT relies on ongoing human-in-the-loop language feedback to iteratively refine predicates and operators. As suggested by its name, it serve as an interpreter between human language feedback and PDDL domain. In contrast, PDDLLM derives predicates in a one-shot, autonomous manner: the LLM summarizes simulated physical interaction rollouts into candidate predicates without requiring human explanations for predicates. This removes the need for continuous human supervision during predicate learning and enables higher autonomy.
>
> ### **Automatic integration with motion planners for our approach:**
> InterPreT requires engineers to predesign motion-planning skills for each primitive action in order to generate executable trajectories (Section III in their paper)[3]. In PDDLLM, the predicates originate from physics-grounded simulations with explicit geometric constraints. This allows us to automatically map symbolic preconditions/effects into motion-planner constraints, eliminating manual motion-planner pre-design.
>
> Rather than competitors, the two methods are complementary: PDDLLM generates the planning domain and integrates motion constraints automatically, while InterPreT can refine the domain by with human feedback.
>
> *We have revised the related work section [L89-95] to clearly reflect the key differences.*
>
> ## **Weakness 2**
> **Our method work with various LLM backbones.** We repeated all experiments with Qwen3-4B and Qwen3-8B. As shown in Table 1, both achieve performance comparable to GPT-4o, with discrepancies within 2–3% and average deviation under 1%. This is expected: once the domain is derived, downstream success largely depends on the PDDL solver rather than the LLM backbone. *We have included a section in Appendix B.21 [L2213] to discuss the use of different LLM backbones, and clarified in the main paper that our approach is not constrained to GPT-4o in Section 6 [L307]*
>
> ### Table R1. Success rate (%) of PDDLLM with Qwen3 series as backbones
>
> |**Task**|**GPT-4o**|**Qwen3-4B**|**Qwen3-8B**|
> |--------|---------|------------|------------|
> |Stack|97.5±1.6|97.7±1.7|97.8±0.3|
> |Unstack|97.7±0.7|97.5±1.6|100±0.0|
> |Color Classification|100±0.0|100±0.0|100±0.0|
> |Alignment|100±0.0|100±0.0|100±0.0|
> |Parts Assembly|100±0.0|100±0.0|98.3±1.1|
> |Rearrangement|64.3±0.7|64.0±1.3|65.1±2.2|
> |Burger Cooking|91.7±4.8|93.1±3.7|95.1±2.5|
> |Bridge Building|87.2±4.3|86.1±3.8|85.0±5.0|
> |Tower of Hanoi|100±0.0|100±0.0|100±0.0|
> |**Overall**|**93.3±0.7**|**93.1±0.4**|**93.6±0.5**|

---

> ### Author Response · Authors · 2025-11-25
> **Reply to W3**
>
> ## **Weakness 3**
> To prevent domain-generation failures from single open-loop prompts, we employ simulation feedback during predicate generation and parallel prompting with execution feedback. These components are not core contribution, so we did not emphasize them in the main text originally. We summarize the technical details as following:
>
> ### **Two feedback loops in the system**
>
> 1. **Simulated Object-Interaction Feedback During Predicate Generation**
> During the predicate generation phase, objects are sampled and simulated to interact with one another. Infeasible interaction (e.g., objects overlapping) or catastrophic failure (e.g., objects fall out of the workspace) are discarded. This feedback filtering ensures that the physical constraints extracted from simulation are accurate and reliable.
>
> 2. **Execution Failure Feedback in Parallel Prompting**
> After receiving the simulation roll out, LLMs are prompted multiple times in parallel to generate planning domain candidates. If a domain encounters a run-time error (syntax, incomplete domain, failure to reach goals,...) it is eliminated from the candidates. If more than one planning domain succeeded, then a majority vote is initiated using the LLM to choose the best one.
>
> **Empirically, the system can steadily generate an executable planning domain when 5 or more parallel prompt are used**. The impact of the number of parallel prompts on the rate of failure to generate executable planning domains is shown in the following table:
>
> #### **Table R2. Failure rate to generate executable domain with varying number of parallel prompts**
>
> |**Tasks**|**1**|**5**|**10**|**15**|**20**|
> |---------|-----|-----|------|------|------|
> |Overall Failure Rate (%)|41.7|8.3|8.3|0|0|
>
> **Additional Demonstrations Help Repair Broken Intermediate Steps** When error encountered in intermediate steps, the system relies on additional demonstrations to repair the domain. In our sub-optimal demonstration experiments(Table 4 [L466], main paper), providing more demonstrations progressively corrected missing or inaccurate predicates and actions, eventually restoring a functional planning domain. Here are some details on this experiment:
>
> Bridge building requires three types of knowledge: picking-up, stacking, and aligning. We initially only provide a picking-up trajectory, result in imprecise domains as the system had no evidence that stacking or aligning mattered, predicates supporting those skills were either discarded or collapsed into vague, weakly constrained forms. When we progressively add demonstrations (first stacking, then aligning) the system began to recognize the relevance of these skills, prioritize their associated predicates, and refine their constraints. The task success rates gradually increased from 0% to over 85%.
>
> *We recognize the importance of more advanced feedback algorithms, thus future work section [L523-525] is revised to discuss the potential combination with advanced feedback approaches. We have also updated figure 1 caption [L179-181], figure 2 caption [L232-234] and section 5.1.3 [L258-265] to clearly describe the two feedback mechanisms used in our method. A section to discuss the impact of parallel prompt is included in Appendix B.16[L2107]. The explanation of additional demonstrations repairing imperfect intermediate steps are revised in [L453-462].*

---

> ### Author Response · Authors · 2025-11-25
> **Reply to Q1-3**
>
> ## **Question1**
> We will address your concerns from the following two aspects:
>
> ### **Handling incorrect constraints:**
>
> 1. **Extensive Simulated Interactions Prevent Catastrophic Errors**
> Catastrophic failures are largely prevented through physics-simulated validation. During predicate generation, each candidate constraint is verified in simulation, and any infeasible or incorrect constraint fails and is discarded, preventing invalid predicates from entering the pool.
>
> 2. **Imprecise predicates can be regenerated with improved demonstration or language guidance.**
> These issue of imprecise predicates can be addressed by **regenerating or re-selecting** from predicate pool using additional demonstrations or clearer task descriptions. We have provided an analysis on predicate regeneration with additional demonstrations in the response to Weakness 3. Here, we include another experiment showing that refining the task description improves predicate precision, with all other settings kept identical to the main paper.
>
> #### **Table R3. Example of predicate refinement with improved language guidance**
>
> | Task Description| Predicates |
> |---------------------|-------------|
> | **Heat Milk** | `(hot-milk ?a): 40 < T < 60 °C`|
> |      | `(warm-milk ?a): 20 < T < 40 °C`  |
> | **Boil Milk** | `(cold-milk ?a): 0 < T < 10 °C`|
> |                    | `(warm-milk ?a): 20 < T < 40 °C` |
> |                     | `(boiling-milk ?a): 90 < T < 100 °C` |
> | **Heat the cold milk from the fridge until it comes to a boil.** | `(cold-milk ?a): 0 < T < 10 °C`  |
> |            | `(near-boil-milk ?a): 80 < T < 90 °C`        |
> |          | `(boiling-milk ?a): 90 < T < 100 °C`   |
>
>
> As task descriptions become more specific, the predicates and their constraints become more precise, as LLMs adjusts the feature-wise hyperparameter $u$ accordingly. The full details on $u$ selection will be provided in our response to Q2.
>
> ### **Generalizable predicate hyperparameters**
> Predicate candidates are generated from simulations that uniformly sample object states, giving broad coverage of possible configurations. As long as test-time scenarios lie within or near this distribution, suitable constraints can be retrieved. We acknowledge that real-world execution may expose corner cases not captured in simulation; in such cases, online constraint generation via robot–environment interaction is a promising future direction [1,2].
>
> *We revise the result section to discuss how the system regenerate and correct predicates in Section 7 [L453-464], and the future work section [L523-525] to discuss future research to integrate environment feedback in our system. The full discussion above is included in Appendix B.18 [L2166]*
>
> ## **Question 2**
> The hyperparameter $u$ is dimension-specific, and we apologize for the confusion caused by the current figure. We will update it to make clear that each feature dimension has its own $u_f$.
>
> **$u_f$ is initialized per feature and later refined by the LLM** For each continuous feature $f$, we initialize $u_f$ using the smallest non-zero observed difference $d_{min}$ of that feature among all relevant objects. This provides a conservative upper bond for $u_f$: it ensures that the learned predicates remain expressive enough to capture distinctions for all objects. During LLM-based predicate generation, the model is allowed to refine the initial discretization step $u_f$ by either merging or subdividing sub-feature intervals. Specifically, the LLM can **merge** intervals that are semantically uninformative for the task into a single broader region, and **subdivide** intervals in feature regions where finer distinctions are necessary. Consequently, even if the initial $u_f$ is imperfect, LLM can recover the task-relevant granularity and generate predicates that remain both expressive and compact.
>
> *We have updated figure 2 [L217] to make clear that each feature dimension has its own value $u_f$. We have revised Section 5.1.4[L267-277] to explain the selection of $u_{f}$*
>
> ## **Question 3**
> We recognize that low-level execution inaccuracies are indeed a common challenge in task and motion planning methods [4,5]. In our framework, we follow the standard task and motion planning paradigm to handle such failures.
>
> **Low-level failure recovery by replanning** When a low-level failure occurs (e.g., the third block falls because the second block was not perfectly centered), it result in a logical state change. The PDDL planner then re-invokes symbolic planning to generate a new sequence of actions that recovers from the failure by re-stacking the misplaced block. This replanning–execution loop is repeated until either the goal is achieved or the time limit is reached. This mechanism ensures that occasional geometric inaccuracies do not terminate the task prematurely.
>
> *We have discussed this task and motion planning feedback loop in the Appendix B.19 [L2187], covering the full discussion above*

---

> ### Author Response · Authors · 2025-11-25
> **Comma Issues & References**
>
> ## **Comma Issues**
>
> Thank you for checking for us. We have revised the corresponding lines[L213-215] to remove comma.
>
>
> ## **References**
> [1] Zhu, et al., “PSALM-V: Automating symbolic planning in interactive visual environments with large language models,” 2025.
>
> [2] Liang et al., “VisualPredicator: Learning abstract world models with Neuro-Symbolic Predicates for robot planning,” 2024.
>
> [3] Han, et al., “InterPreT: Interactive predicate learning from language feedback for generalizable task planning,” 2024.
>
> [4] Silver, et al., “Planning with learned object importance in large problem instances using graph neural networks,” 2021.
>
> [5] Toussaint, “Logic-geometric programming: an optimization-based approach to combined task and motion planning,” 2015.
>
> [6] Black et al., π_0: A Vision-Language-Action Flow Model for General Robot Control, 2024

---

### Official Review · Reviewer_3sZG · 2025-11-01

**Soundness:** 3
**Presentation:** 3
**Contribution:** 3
**Rating:** 6
**Confidence:** 2

**Summary:**

The submission proposes a framework to derive a symbolic planning domain from a single demonstration. It achieves this through predicate “imagination” from simulator roll-outs summarized by an LLM and action “invention” from logical state transitions. Then the produced domain can be used with a PDDL motion planner. The authors evaluate across  several tasks and environments and report large gains over LLM-enabled planners (with a fixed planning budget), includingl real-robot executions on three platforms.

**Strengths:**

The submission tackles the costly manual domain-spec bottleneck in TAMP and positions the work among LLM planners and domain-inference lines of work. The end-to-end automation pipeline (predicate imagination, action invention  and LoCA) is, in my understanding, the main contribution/novelty. The tasks are varied in difficulty and nature (Tower of Hanoi, bridge building, burger cooking), and multiple SOTA baselines are included (LLMTAMP, LLMTAMP-FF/FR, o1-TAMP, R1-TAMP, RuleAsMem). Analysis of time-limit is provided, which can be relevant when planning under a fixed time budget in real applications. Real robot demonstrations strengthen the empirical support of the frameworks effectiveness.

**Weaknesses:**

Since the pipeline still relies on some hand-chosen design choices (e.g. $u$ for subspace granularity), statements about constructing domains "without manual predesign" might be exaggerated.

How often does the limited operator set miss required invariants? First-order predicates come from discretized subspaces while higher-order ones use a limited set of logical operators/quantifiers. The limitations section admits missing complex predicates (e.g., ordering constraints), which can materially affect plans in more challenging domains.

**Questions:**

Can you add an ablation for u and for the number of parallel prompts? Including domain quality metrics (missing/redundant predicates)

Can you include more quantitative per platform metrics for real robot experiments?

Can you disambiguate token costs across domain derivation and per-task planning?

---

> ### Author Response · Authors · 2025-11-25
> **Reply to W1&2**
>
> Thank you for recognizing the contribution of our end-to-end domain-construction pipeline and the breadth of our empirical evaluation. We appreciate the insightful questions raised, they help us to better clarify our method. We have revised the paper based on your feedback and highlighted all changes with the corresponding line-number references. we would like to address your concerns in detail below.
>
> ## **Weakness 1**
> **Statement Adjustment from 'Without' to 'Minimal'** : We intended to say human labor would not directly required for planning domain engineering but the original wording may have been confusing and too strong. We will revise the claim to more accurately reflect the degree of automation in our pipeline. We wonder if you feel this alternative would be proper: construct domains with minimal manual effort. *We have updated the abstract [L21] and introduction section [L52-54] in the revised version accordingly.*
>
>
>
> ## **Weakness 2**
> **The Selection of the Operator Set**： We acknowledge that using more operators could yield more complete predicate candidates, but it would also expand the search space combinatorically, making it harder for the LLM to select the useful ones. We therefore adopt a compact operator set empirically validated effective in prior work [1,2]. *We have revised the paper in Section 5 [L207-209] to explain the selection of these operators.*
>
> **Frequency of Missing Complex Predicates Depends on Their Logical Complexity** These complex predicates are missed more frequently when they require deep or repeated operator compositions (e.g., `all_base_finished`, involving nested *forall*). Such structures enlarge the candidate pool, making it harder for the LLM to identify them reliably. In contrast, simpler higher-order predicates, such as `on_top`, involving shallow compositions, are generated reliably with almost no missing observed.
>
>
> **Missing Complex Predicates Does Not Cause Immediate Failure, but We Agree This is a Limitation Under a Time Budget.**
> Missing highly composite predicates rarely causes direct failure. The essential geometric and relational structure is already captured by first-order predicates; higher-order ones mainly speed up planning by guiding action order. For example, omitting `all_bases_finished` in bridge building may cause premature surface assembly and trigger a backtrack, resulting longer planning time. We acknowledge that this becomes more problematic for time-sensitive tasks.
> *Thus, we have revised the limitation discussion section [L504-512] in the main paper to clarify the consequence of missing complex predicates, the future work section [L523-525] to discuss the possible work of extending predicates from failure feedback, and have included the detailed analysis above in Appendix B.17 [L2148].*

---

> ### Author Response · Authors · 2025-11-25
> **Reply to Q1**
>
> ## **Question 1**
> We conducted ablation studies examining the impact of the number of parallel prompts $n_{\text{prompt}}$ and the feature subspace granularity parameter $u_f$. These experiments extend the result of Table 3 in main paper and cover four representative tasks: stacking, Tower of Hanoi, bridge building, and burger cooking. Empirically, when $n_{prompt}>=5$ the system can steadily generate an executable planning domain.
>
>
> ### **Ablation of $u_f$**
> For subspace granularity, we perform an ablation study on the selection of $u$. For each continuous feature $f$, $u_f$ has to be smaller than the smallest non-zero observed difference $d_{min}$ of that feature among all relevant objects. This provides a conservative upper bond for $u_f$. Otherwise, the system cannot capture the distinctions of all objects. Thus, we ablate the selection of u from $d_{min}$ to $0.25d_{min}$. We fix $n_{prompt}=10$ for all experiments. Three runs were performed for each $u_f$ per task, and the average missing/redundant percentage is reported.
>
> #### **Table R1. Effect of the hyperparameter $u_f$ on redundant and missing predicates percentages**
>
> | **Tasks** |  | **$u = d_{\min}$** | **$u = 0.75d_{\min}$** | **$u = 0.5d_{\min}$** | **$u = 0.25d_{\min}$** |
> |----------|--|----------------------|---------------------------|---------------------------|----------------------------|
> | **Stacking** | Redundant | 8.3% | 12.5% | 16.7% | 12.5% |
> |            | Missing   | 4.2% | 0.0%  | 0.0%  | 4.2% |
> | **Tower of Hanoi** | Redundant | 14.3% | 14.3% | 9.5% | 14.3% |
> |                 | Missing   | 0.0% | 0.0% | 0.0% | 4.2% |
> | **Bridge Building** | Redundant | 3.7% | 11.1% | 14.8% | 22.2% |
> |                    | Missing   | 22.2% | 22.2% | 22.2% | 22.2% |
> | **Burger Cooking** | Redundant | 3.7% | 7.4% | 18.5% | 22.2% |
> |                   | Missing   | 22.2% | 22.2% | 25.9% | 22.2% |
> | **Overall** | Redundant | 7.5% | 11.3% | 14.9% | 17.8% |
> |             | Missing   | 12.1% | 11.1% | 12.0% | 13.2% |
>
> **To conclude, our method is relatively stable with changing $u$.** This is expected because the LLM can adapt the discretization during predicate generation: it may merge intervals that are semantically uninformative for the task and subdivide intervals in regions where finer distinctions are required.  Analyzing the trend, smaller $u$ would increase the risk of redundant predicates, but the negative effect on missing predicate is less apparent.
>
>
> ### **Ablation of $n_{prompt}$**
>
> The result of varying $n_{prompt}$ is shown in the table below. We fix $u_f=d_{min}$ for all experiments. Three runs were performed for per $n_{prompt}$ per task, and the average missing/redundant percentage is reported.
>
> #### **Table R2. Redundant and missing predicate rates across tasks as the number of parallel prompts changes.**
> | **Tasks** | **Type** | **1** | **5** | **10** | **15** | **20** |
> |-----------|----------|-------|-------|--------|--------|--------|
> | **Stacking** | Redundant | 25.0% | 12.5% | 8.3% | 4.2% | 4.2% |
> |  | Missing | 4.2% | 0.0% | 4.2% | 0.0% | 0.0% |
> | **Tower of Hanoi** | Redundant | 38.1% | 9.5% | 14.3% | 14.3% | 9.5% |
> |  | Missing | 0.0% | 0.0% | 0.0% | 0.0% | 0.0% |
> | **Bridge Building** | Redundant | 33.3% | 11.1% | 3.7% | 3.7% | 0.0% |
> |  | Missing | 29.6% | 25.9% | 22.2% | 22.2% | 22.2% |
> | **Burger Cooking** | Redundant | 29.6% | 11.1% | 3.7% | 11.1% | 0.0% |
> |  | Missing | 29.6% | 22.2% | 22.2% | 22.2% | 22.2% |
> | **Overall** | Redundant | 31.5% | 11.1% | 7.5% | 8.3% | 3.4% |
> |  | Missing | 15.8% | 12.0% | 12.1% | 11.1% | 11.1% |
>
> The table indicates that increasing $n_{prompt}$ is generally helpful in eliminating the redundant predicates, but does little to help in producing missing predicates. This is because the missing predicates like `all_base_finished` are highly complex in logical reasoning and physics grounding, which go beyond the ability of the LLM backbone.
>
> *We describe our method for selecting $u_f$ in Section 5.1.4 (L267-277). Our experimental settings are specified in Section6 (L306-308). We have also updated Table 3 (L466-473) to report the average over multiple independent runs. The full ablation studies are provided in Appendix B.15 (L2070) and Appendix B.16 (L2102).*

---

> ### Author Response · Authors · 2025-11-25
> **Reply to Q2, Q3, & References**
>
> ## **Question 2**
> We conducted 10 runs of each real robot experiment described in Section 8. The Tower of Hanoi, bridge-building, and burger-making experiments are executed using the original PDDLLM system. The table-top stacking task is performed using PDDLLM with VLA(Pi0[6]) as the low-level manipulation module; the technical details are provided in Appendix B.1.3 [L1026] The statistics are obtained as below:
>
> #### **Table R3. Real-world success rate across tasks**
>
> | **Task**                           | **Success Rate** |
> |-----------------------------------|------------------|
> | Tower of Hanoi on Franka Panda    | 9/10             |
> | Bridge Building on Franka Panda   | 8/10             |
> | Burger Making on Agilex Piper     | 7/10             |
> | Object Stacking on UR5e           | 7/10             |
>
> *We have included the above table summarizing the performance of our approach on different robot platforms in Appendix B.1 (L822).*
>
>
> ## **Question 3**
> Our method only queries the LLM in the planning domain derivation phase. In the task planning phase, it relies on search-based PDDL solvers. Thus, the token cost comes from the domain derivation phase. This makes it well suited for long-term deployments that repeatedly perform similar tasks. The following table shows the token cost observed in each of 5 trials for three tasks. The average value is reported in the main paper. Each trail we adopt $u = d_{\min}$ and $n_{prompt} = 10$. *We revise the paper to include Table R2 in Appendix B.20 [L2199] and modified Section 7 [L437-439] to disambiguate token costs across domain derivation and per-task planning*
>
> #### **Table R4. Token cost (k) across 5 trials for each task**
>
> | **Task**            | **Trial 1** | **Trial 2** | **Trial 3** | **Trial 4** | **Trial 5** | **Avg** |
> |---------------------|-------------|-------------|-------------|-------------|-------------|---------|
> | Rearrangement       | 306.4       | 292.9       | 322.5       | 465.1       | 285.2       | 334.4   |
> | Tower of Hanoi      | 478.6       | 526.6       | 558.6       | 547.2       | 564.1       | 535.1   |
> | Bridge Building     | 293.0       | 478.2       | 336.5       | 471.0       | 298.0       | 375.3   |
>
> ### **References**
> [1] Silver et al., “Predicate Invention for Bilevel Planning,” 2023.
> [2] Chitnis et al., “Learning neuro-symbolic relational transition models for bilevel planning,” 2022.

---

### Official Review · Reviewer_h5Ug · 2025-11-01

**Soundness:** 3
**Presentation:** 3
**Contribution:** 3
**Rating:** 6
**Confidence:** 4

**Summary:**

This paper proposes PDDLLM, a framework that automatically generates the PDDL planning domain from one demonstration by combining LLMs with physical simulation. It introduces a Logical Constraint Adapter (LoCA) to automatically ground the generated symbolic actions into motion constraints, enabling seamless integration with motion planners and real-robot execution. Experiments on 9 tasks show that PDDLLM outperforms baselines and achieves performance comparable to expert-designed domains.

**Strengths:**

1. This paper aims to automatically generate planning domains from one demonstration to reduce manual engineering efforts, which is a valuable goal for the field.
2. Experiments on 9 tasks show that the proposed method achieves a high success rate, outperforming other baselines.

**Weaknesses:**

1. The paper relies on a physics simulator to evaluate the physical feasibility of predicates. However, such simulation-based evaluation may fail to capture complex dynamics, limiting the method’s generalization to real-world settings. The current experiments only involve simple rigid-body interactions, so it remains unclear how the proposed approach would perform with more complex objects such as deformable materials or fluids.
2. There are several unclear aspects in the paper:
(1) It is not clearly stated whether the predicates must be predefined or can be freely generated by the LLM based on the task and scene. If predicates are required in advance, this could constrain the generality of the proposed framework.
(2) The paper mentions that “the range of each feature is divided into intervals, with the length of each interval being a hyperparameter,” but does not specify how this hyperparameter is determined. Is it fixed manually or generated adaptively by the LLM?

**Questions:**

The paper claims that the framework can integrate knowledge across demonstrations. In this context, during the evaluation, were demonstrations from other tasks used to assist the construction of the domain model for the current task?

---

> ### Author Response · Authors · 2025-11-25
> **Reply to W1**
>
> Thank you for highlighting the value of our goal, our novelty and the strong empirical results. We appreciate the thoughtful questions raised, they help us to better clarify our method. We have revised the paper based on your feedback and highlighted all changes with the corresponding line-number references. We address each of them carefully below.
>
> ## **Weakness 1**
>
> We acknowledge that domains involving complex geometries and dynamics, such as deformable objects or fluid interactions, are still difficult to handle reliably in simulation, both for our system and for the wider robotics community [1–6]. In response, we provide a discussion of the scenarios where our method is suitable, and show how it can still offer practical benefits even when the environment exhibits complex dynamics.
>
> **Suitable Situations**
> Our approach is most effective in domains that align with the classical task-and-motion-planning research scope [1,4] with (1) consistent object physical attributes, (2) scene geometry can be represented with structured models such as URDF/CAD, so that motion-planning constraints can be grounded reliably.
>
> Prior work shows that those suitable situations cover a meaningful range of common robotic manipulation scenarios, including table tidying, object sorting, cabinet and drawer organization, assembly, logistics. etc [1–6].
>
> ### **Our method remain helpful in challenging domains with complex dynamics**
> Our method can help are two main aspects:
> 1. **Easing Human Workload by Automating Common Dynamics Predicate Generation** In challenging settings, the generated domain captures the common dynamics, leaving only a few dynamically complex predicates for engineers to refine instead of rebuilding the entire domain.
> 2. **Our method works well with algorithms designed for complex geometry and dynamics.** Our approach can be paired with stronger low-level manipulation modules such as VLA to handle complex dynamics and geometry. *To illustrate this, we have included an experiment using a finetuned version of Pi0[7] as the low-level skill controller, achieving 7/10 success rate in real-robot experiment. The experiments are presented in Section 8[L482-485], with additional technical details provided in Appendix B.1.3. [L1026]*
>
> *We have first updated the limitations [L512-513] to mention this constrain, and add future work section[L523-525] to propose direct-interaction with real environment as an future direction. A section in Appendix B.14[L2037] includes the full discussion above.*

---

> ### Author Response · Authors · 2025-11-25
> **Reply to W2**
>
> ## **Weakness 2**
>
>
> ### **LLMs are Free to Generate Predicate**
> Predicates in our framework do not need to be predefined. The system is able to generate new predicates through summarizing the simulation roll-out using the LLM when presented with novel tasks or scenes. As additional illustration, **we provide 8 additional examples of predicates generated by our system under different tasks and scenes in Appendix B9[L1836]**, we hope this would help to better demonstrate our method. *We have also revised the introduction section to clarify the predicates are generated by LLM[L52-54]. A few representative tasks and associated predicate generated by LLM is shown in the following table.*
>
> #### **Table R1. Examples of predicates for different tasks and scenes**
>
> | **Task**                          | **Predicate Example**                            |
> |-----------------------------------|--------------------------------------------------|
> | **Robot arm lifts a box in warehouse** | `(light-box ?a): 40 < weight < 70 kg`          |
> |                                   | `(heavy-box ?a): 70 < weight < 100 kg`          |
> | **Fulfill a 15-cm coffee cup**   | `(needs-refill ?a): 0 < level < 7.5 cm`         |
> |                                   | `(nearly-full ?a): 7.5 < level < 11.25 cm`      |
> | **Boil milk**                    | `(cold-milk ?a): 0 < T < 10 °C`                 |
> |                                   | `(warm-milk ?a): 20 < T < 40 °C`                |
> |                                   | `(boiling-milk ?a): 90 < T < 100 °C`            |
> | **Fix broken light in office**   | `(dimly-lit ?a): 0 < brightness < 150 lux`      |
>
>
>
> ### **$u_f$ is manually initialized per feature and later refined by the LLM**
> For each continuous feature $f$, we initialize $u_f$ using the smallest non-zero observed difference $d_{min}$ of that feature among all relevant objects. This provides a conservative upper bond for $u_f$: it ensures that the learned predicates remain expressive enough to capture distinctions for all objects.
> During LLM-based predicate generation, the model is allowed to refine the initial discretization step $u_f$ by either merging or subdividing sub-feature intervals. Specifically, the LLM can **merge** intervals that are semantically uninformative for the task into a single broader region, and **subdivide** intervals in feature regions where finer distinctions are necessary. Consequently, even if the initial $u_f$ is imperfect, LLM can recover the task-relevant granularity and generate predicates that remain both expressive and compact. Below we present an ablation study on $u_f$ to examine its affects the quality of the generated planning domain, measured by the percentage of missing and redundant predicates, relative to expert-designed domains.
>
> #### **Table R2. Effect of the hyperparameter $u_f$ on redundant and missing predicates percentages**
>
> | **Tasks** |  | **$u = d_{\min}$** | **$u = 0.75d_{\min}$** | **$u = 0.5d_{\min}$** | **$u = 0.25d_{\min}$** |
> |----------|--|----------------------|---------------------------|---------------------------|----------------------------|
> | **Stacking** | Redundant | 8.3% | 12.5% | 16.7% | 12.5% |
> |            | Missing   | 4.2% | 0.0%  | 0.0%  | 4.2% |
> | **Tower of Hanoi** | Redundant | 14.3% | 14.3% | 9.5% | 14.3% |
> |                 | Missing   | 0.0% | 0.0% | 0.0% | 4.2% |
> | **Bridge Building** | Redundant | 3.7% | 11.1% | 14.8% | 22.2% |
> |                    | Missing   | 22.2% | 22.2% | 22.2% | 22.2% |
> | **Burger Cooking** | Redundant | 3.7% | 7.4% | 18.5% | 22.2% |
> |                   | Missing   | 22.2% | 22.2% | 25.9% | 22.2% |
> | **Overall** | Redundant | 7.5% | 11.3% | 14.9% | 17.8% |
> |             | Missing   | 12.1% | 11.1% | 12.0% | 13.2% |
>
> **To conclude, our method is relatively stable with changing $u$.** This is expected because the LLM can adapt the discretization during predicate generation. Analyzing the trend, smaller $u$ would increase the risk of redundant predicates, but the negative effect on missing predicate is less apparent.
>
> *We have clarified the procedure for selecting hyperparameters in Section 5.1.4 [L267-277] of the updated manuscript and included the above ablation study in Appendix B.15 [L2070].*

---

> ### Author Response · Authors · 2025-11-25
> **Reply to Q1 & References**
>
> ## **Question 1**
> **We show that our method performs well both with and without assistance from demonstrations of other tasks.**  In the main evaluation (Table 2, Section 7), PDDLLM by default uses only a single corresponding demonstration per task, without access to demonstrations from other tasks. To assess its ability to integrate cross-task knowledge, we also evaluated settings where only demonstrations from other simpler related tasks were provided for a more complex current task. Specifically, for the rearrangement, the model received only demonstrations of stacking and unstacking, while for the bridge building, it was given only demonstrations of stacking and alignment.
>
> In the ablation study on robustness to suboptimal demonstrations (Table 4, Section 7), we also allow the system to incorporate and aggregate information across multiple sub-optimal demonstrations to repair missing knowledge and construct a complete planning domain.
>
> *In the Section 6 [L310-314] of revised manuscript, we have updated the experiment set-up description to make experiment design explicit. We also include the above discussion in Section 7 [L453-461] to highlight the system’s capability to combine multiple demonstrations for domain generation.*
>
> ### **References**
>
> [1] Silver, et al., “Planning with learned object importance in large problem instances using graph neural networks,” 2021.
>
> [2] Toussaint, “Logic-geometric programming: an optimization-based approach to combined task and motion planning,” 2015.
>
> [3] Kumar et al., “Learning Efficient Abstract Planning Models that Choose What to Predict,” 2023.
>
> [4] Garrett et al., “Integrated Task and Motion Planning,” ,2021.
>
> [5] Kase, et al., “Transferable task execution from pixels through deep planning domain learning,” 2020.
>
> [6] Liang et al., “VisualPredicator: Learning abstract world models with Neuro-Symbolic Predicates for robot planning,” 2024.
>
> [7] Black et al., π_0: A Vision-Language-Action Flow Model for General Robot Control, 2024

---

### Official Review · Reviewer_MY9K · 2025-11-01

**Soundness:** 3
**Presentation:** 2
**Contribution:** 2
**Rating:** 4
**Confidence:** 3

**Summary:**

This paper presents a new method that is capable to generate a complete planning domain from scratch, without relying on any predefined predicates or actions. The method employs the reasoning ability of LLMs to do generation based on demonstration.

**Strengths:**

This work targets an interesting and meaningful problem in planning.

The proposed methodology does not rely on pre-defined predicate space and action model, which reduce the effort of human annotation.

Experimental in real robot environments demonstrates the effectiveness of the proposed method.

**Weaknesses:**

**Major**

Successful deployment of the proposed method requires a perception function that can accurately extract the continuous states from objects. It is unclear what types of the perception function the proposed method can work along well.

It was not extensively discussed in the paper how robust the proposed method is with respect to any noises in the perception process.

It seems that applying the proposed method in real applications requires setting up a same digital copy in a simulation. This limits the potential applicable areas as setting up simulation in some domains requires laborious manual modelling even for feature parameters. It would be better to discuss what kind of domain the proposed method can easily handle, and what domains the proposed method may encounter big challenges.

**Minor**

Some paragraph is not written with clear motivations, which makes the reading not easy to follow. For example, L180-L196, it is unclear why the method is dividing the feature space. It would be better to clearly explain the target problem at the moment, e.g., use parallel simulations to imagine predicates by summarizing the simulation roll-outs with LLMs, and explain what the challenges are.

**Questions:**

Please see the weaknesses part.

---

> ### Author Response · Authors · 2025-11-25
> **Reply to W1&2**
>
> Thank you very much for recognizing the importance of our problem setting, the novelty of removing reliance on pre-defined symbolic structures, and the effectiveness demonstrated in our real-robot experiments. We also appreciate the constructive feedback, which has helped us improve paper presentation and clarify our approach. We have revised the paper based on your feedback and highlighted all changes with the corresponding line-number references. Below, we address each concern in detail.
>
> ## **Weakness 1 & 2**
> We acknowledge that the requirement for decent perception is a general and well-known open challenge shared by work in task and motion planning [1-4]. Here, we would like to discuss the compatibility of our methods to perception module.
>
> ### **Perception in our real-robot system**
>
> For our real-world experiments, we adopt ArUco markers for perception, which provides reliable object identity and 6D pose estimates, enabling us to evaluate the planning framework without significant perception errors.
>
> ### **Compatibility with other perception modules**
>
> PDDL task planning is proven modular and can integrate with various perception systems, such as classic 6D pose estimation[3], learning based methods[5], or VLMs[6]. As our method follows the standard PDDL pipeline, it imposes no restrictions on how object states are obtained.
>
> ### **Robustness to perception noise**
>
> Our method represents logical abstractions as intersections of intervals, which inherently tolerates perception errors by operating over value ranges rather than exact measurements. To further evaluate robustness to perception noise, we conducted an additional real-world experiment in which two 4 cm × 4 cm × 2 cm cubes were randomly placed on a table, their 6D poses estimated using our ArUco system, and predicate values compared against ground-truth spatial relationships. For each predicate (`is_on` and `next_to` ), we generated **50 balanced trials**, 25 true, 25 false.
>
> #### **Table R1. Real-word Predicate Accuracy of PDDLLM**
> | **Predicates** | `is_on` | `next_to` |
> |----------------|---------|-----------|
> | Accuracy | 94% | 96% |
>
> We acknowledge that pose estimation accuracy may vary across different perception methods. To evaluate robustness under less reliable perception, we conducted an additional experiment by injecting percentage noise ranging from 5\% to 30\% into the perceived object states. Each setting was repeated three times, yielding the following results:
>
> #### **Table R2. Predicate Accuracy under Injected Percentage Perception Error**
> | **Percentage Noise** | **5%**         | **10%**        | **15%**        | **20%**        | **25%**        | **30%**        |
> |----------------------|----------------|----------------|----------------|----------------|----------------|----------------|
> | **is_on Accuracy**   | 92.7 ± 0.9     | 88.7 ± 2.5     | 84.0 ± 3.3     | 80.7 ± 3.4     | 80.7 ± 1.9     | 78.7 ± 0.9     |
> | **next_to Accuracy** | 94.7 ± 0.9     | 84.0 ± 1.6     | 80.0 ± 4.9     | 76.0 ± 4.3     | 74.0 ± 2.8     | 69.3 ± 1.9     |
> | **Overall Accuracy** | 93.7 ± 1.4     | 86.3 ± 3.1     | 82.0 ± 4.6     | 78.3 ± 4.5     | 77.3 ± 4.1     | 74.0 ± 4.9     |
>
>
>
> The experiment shows our approach has decent tolerance to perception noise, with clear degradation only beyond 20\% noise. This indicates that the system can operate reliably with typical perception accuracy.
>
> We also acknowledge that extending the framework to handle richer visual inputs and more challenging perception conditions is a promising direction for future work. *We have added a new section in Appendix B.13 (L1995) discussing the impact of perception noise, including the two experiments described above. We also updated the limitation section (L512–514) to clarify this constraint and revised the future work section (L519–523) to outline relevant research directions.*

---

> ### Author Response · Authors · 2025-11-25
> **Reply to W3&4 and References**
>
> ## **Weakness 3**
> We agree that simulation, while helpful, can be time-consuming to configure and may struggle in complex environments. To address this, we clarify the types of settings where our method works reliably and highlight the domains where it may face challenges. We also provide evidence on how our method can remain helpful in the challenging domains.
>
> ### **Our method can easily handle domains aligned with traditional task and motion planning (TAMP) setting**
>
> The proposed system is naturally suited to domains that align with the classical TAMP setting [1,4], namely those with: (1) consistent object physical attributes, (2) scene geometry can be represented with structured models such as URDF/CAD, so that motion-planning constraints can be grounded reliably.
>
> Prior work shows that those suitable situations cover a meaningful range of common robotic manipulation scenarios, including table tidying, object sorting, cabinet and drawer organization, assembly, logistics. etc [1–6]. These simulation environments are widely studied in robotics, and mature toolkits offer many ready-to-use environments (Pybullet, Issac sim, Robosuite,  ect.).
>
> ### **What domains are challenging**
>
> Despite the strength of modern simulators, certain domains remain a challenge to us, and to the robotic planning community:
>
> 1. Deformable, fluid, or highly non-rigid materials, where geometry definition is under studied and simulation is limited.
> 2. Tasks requiring fine-grained feature modeling, such as dexterous geometric manipulation, where small simulation errors yield large discrepancies in behavior
> 3. Unstructured or rapidly changing environments, where object properties vary unpredictably
>
> ### **Our method remain helpful in challenging domains**
> There are two main aspects:
> 1. **Easing Human Workload by Automating Common Dynamics Predicate Generation** In challenging settings, the generated domain captures the common geometry and dynamics, leaving only a few complex predicates for engineers to refine instead of rebuilding the entire domain.
> 2. **Our method works well with algorithms designed for complex geometry and dynamics.** Our approach can be paired with stronger low-level manipulation modules such as VLA to handle complex dynamics and geometry. *To illustrate this, we have included an experiment using a finetuned version of Pi0[7] as the low-level skill controller, achieving 7/10 success rate in real-robot experiment. The experiments are presented in Section 8[L482-485], with additional technical details provided in Appendix B.1.3 [L1026]*
>
> ### **Partial simulation can reduce engineering overhead**
>
> Although our method relies on simulation, a full digital copy of the environment is often unnecessary. In many cases, it suffices to model only local object-specific interactions rather than reconstruct the entire scene. For example, modelling only two object to learn `is_on` instead of the whole room.
>
> *We have first updated the limitations [L512-514] to mention this constrain, and add future work section[L523-525] to propose direct-interact with real environment as an future direction. A section in Appendix B.14[L2036] includes the full discussion above*
>
> ## **Weakness 4**
> Here are the main motivations behind our discretization step:
>
> (1) **Helping the LLM identify meaningful feature ranges.**
> Raw continuous simulation samples scatter across the feature space, making it difficult for the LLM to detect consistent patterns or determine meaningful numeric thresholds. Discretization groups similar states into coherent regions, enabling the LLM to summarize the dominant relational patterns within each region.
>
> (2) **Providing stable physical boundaries for downstream planning.**
> The interval boundaries derived from simulations become physically grounded numeric constraints. These stable boundaries allow the motion planner to reliably evaluate predicate truth values during planning.
>
> *We have revised [L192-196] to clarify these motivations. We also carefully reviewed the entire methodology section to ensure design motivation is clearly explained, including explaining motivation to combine LLM and simulation [L143-147] and motivation for higher-order predicates [L207-209].*
>
> ## **References**
> [1] Silver, et al., “Planning with learned object importance in large problem instances using graph neural networks,” 2021.
>
> [2] Toussaint, “Logic-geometric programming: an optimization-based approach to combined task and motion planning,” 2015.
>
> [3] Kumar et al., “Learning Efficient Abstract Planning Models that Choose What to Predict,” 2023.
>
> [4] Garrett et al., “Integrated Task and Motion Planning,” ,2021.
>
> [5] Kase, et al., “Transferable task execution from pixels through deep planning domain learning,” 2020.
>
> [6] Liang et al., “VisualPredicator: Learning abstract world models with Neuro-Symbolic Predicates for robot planning,” 2024.
>
> [7] Black et al., π_0: A Vision-Language-Action Flow Model for General Robot Control, 2024

---

### Author Response · Authors · 2025-12-01
**Rebuttal Summary**

We deeply appreciate the Area Chair’s careful coordination of the review process, and we thank all Reviewers for acknowledging the novelty of our approach and for providing feedback. To assist the AC's assessment, we provide a concise summary of our responses and revisions below.

**Overall Assessment** All Reviewers recognized the novelty of our pipeline for automating planning-domain generation and were impressed by the strong empirical results in both simulation and real-robot evaluations. Additionally, reviewers **MY9K** and **h5Ug** emphasized that the problem addressed is practically meaningful for robotic planning. Reviewers **3sZG** and **ZvYq** highlighted the strength of our end-to-end design. Reviewer **3sZG** noted the breadth of our baseline comparisons, and Reviewer **ZvYq** underscored our method’s strength to transfer domain knowledge to unseen tasks and environments.

Importantly, no reviews pointed to factual inaccuracies or fundamental methodological issues. The concerns are mainly requests for expanded validation (additional backbone models, hyperparameter ablation studies) and clarifications (compatibility with different perception models, failure recovery, and applicable scenarios). We have comprehensively addressed all these points by detailed responses and revisions to the manuscript accompanied by added experimental results.

Specifically, we addressed the following:

**Ablation Study on Hyperparameters** (Reviewers h5Ug, 3sZG, ZvYq) We added ablation studies showing that our method is stable with respect to the discretization scale `u_f`. The LLM-driven automated adaptation compensates for imperfect initial values. Increasing `n_prompt` improves domain quality until reaching a plateau. *(Appendix B.15 and B.16)*

**Performance with Other LLM Backbones** (Reviewer ZvYq) We replicated our method using the Qwen3 series (4B and 8B) in place of GPT-4o and observed comparable performance. *(Appendix B.21)*

**Perception Module Compatibility** (Reviewer MY9K)
Our response clarifies that our approach inherits PDDL’s compatibility with diverse perception modules. We also added experiments demonstrating robustness to perception noise. *(Appendix B.13)*

**Failure Recovery** (Reviewer ZvYq) We added experiments showing that additional demonstrations and refined language instructions effectively correct imprecise domain generation. *(Lines 453–464 and Appendix B.18)*

**Applicable Scenarios** (Reviewers MY9K, h5Ug) We clarified that our work is situated within standard robot task and motion planning. Scenarios involving complex dynamics (such as fluids or deformable objects) that cannot currently be reliably simulated fall outside our scope. Nevertheless, we explain how our method can still offer utility even in such settings. *(Appendix B.14)*

**Summary** Overall, the reviewers’ reception of our work was positive, with several strengths identified. We appreciate the reviewers’ constructive criticism and their requests for clarification and additional experiments. These have significantly strengthened the paper. We are confident that the revisions and new results address the reviewers’ main concerns, and we believe they would have acknowledged this had the review process proceeded as planned.

---

### Meta-Review · Area_Chair_gZeS · 2026-01-08

**Summary:**

This work proposes PDDLLM, a method that uses an LLM for the automatic extraction of a symbolic planning domain from a single demonstration. Overall, the reviewers are positive about the work, and the rebuttal addresses most of the major concerns. The AC agrees with the authors that concerns regarding perception issues are generally applicable to the wider field of TAMP. Hence, the AC recommends acceptance. The authors are encouraged to revise the paper following the feedback from the reviewers.

**Reviewer Concerns:**

Adressed:
- Perception Requirements
- Simulation Requirements
- Hyperparameter Selection
- Claims of "Without Manual Predesign" May Be Exaggerated - the author acknowledged to tone it down

**Reviewer Scores:**

Reviewer MY9K -> likely increase from 4 to 6
Other scores likely remain the same.

---

### Decision · Program_Chairs · 2026-01-26

Accept (Poster)